# Sensitivity of the southern hemisphere circumpolar jet response to Antarctic ozone depletion: prescribed versus interactive chemistry

Sabine Haase[1], Jaika Fricke[1], Tim Kruschke[2], Sebastian Wahl[1], and Katja Matthes[1,3]

[1]GEOMAR Helmholtz Center for Ocean Research Kiel, Kiel, Germany
[2]Swedish Meteorological and Hydrological Institute - Rossby Centre, Norrköping, Sweden
[3]Christian-Albrechts-Universität zu Kiel, Kiel, Germany

**Correspondence:** Sabine Haase (shaase@geomar.de)

**Abstract.** Southern hemisphere lower stratospheric ozone depletion has been shown to lead to a poleward shift of the tropospheric jet stream during austral summer, influencing surface atmosphere and ocean conditions, such as surface temperatures and sea ice extent. The characteristics of stratospheric and tropospheric responses to ozone depletion, however, differ among climate models depending on the representation of ozone in the models.

The most appropriate way to represent ozone in a model is to calculate it interactively. However, due to computational costs, in particular for long–term coupled ocean–atmosphere model integrations, the more common way is to prescribe ozone from observations or calculated model fields. Here, we investigate the difference between an interactive and a specified chemistry version of the same atmospheric model in a fully–coupled setup using a 9–member chemistry–climate model ensemble. In the specified chemistry version of the model the ozone fields are prescribed using the output from the interactive chemistry

model version. We use daily–resolved ozone fields in the specified chemistry simulations to achieve a very good comparability between the ozone forcing with and without interactive chemistry. We find that although the short–wave heating rate trend in response to ozone depletion is the same in the different chemistry settings, the interactive chemistry ensemble shows a stronger trend in polar cap stratospheric temperatures (by about $0.7 \, \mathrm{K} \, \mathrm{decade}^{-1}$) and circumpolar stratospheric zonal mean zonal winds (by about $1.6 \, \mathrm{ms}^{-1} \, \mathrm{decade}^{-1}$) as compared to the specified chemistry ensemble. This difference between interactive and spec-

ified chemistry in the stratospheric response to ozone depletion also affects the tropospheric response. However, an impact on the poleward shift of the tropospheric jet stream is not detected.

We attribute part of the differences found in the experiments to the missing representation of feedbacks between chemistry and dynamics in the specified chemistry ensemble, which affect the dynamical heating rates, and part of it to the lack of spatial asymmetries in the prescribed ozone fields. This effect is investigated using a sensitivity ensemble that was forced by a three–

dimensional instead of a two–dimensional ozone field.

This study emphasizes the value of interactive chemistry for the representation of the southern hemisphere stratospheric jet response to ozone depletion and infers that for periods with strong ozone variability (trends) the details of the ozone forcing could also have an influence on the representation of southern hemispheric climate variability.

## 1 Introduction

The last two decades of the $20^{th}$ century were characterized by a strong loss in polar lower stratospheric ozone during spring through catalytic heterogeneous chemical processes involving anthropogenically released halogenated compounds, such as those including chlorine and bromine (Solomon et al., 2014). Ozone depletion was especially strong in the southern hemisphere (SH) due to more favourable environmental conditions, i.e. a very stable, strong and cold polar stratospheric vortex.

The annually reoccurring depletion in polar stratospheric ozone was striking. Political action was begun that ultimately led to a ban on the responsible substances (termed: ozone depleting substances, ODSs) under the Montreal Protocol in 1987. Nevertheless, due to their long lifetimes, ODSs still influence chemistry and radiation balances in the atmosphere and SH spring ozone concentrations will remain low until the middle of the $21^{st}$ century. Latest simulations from the Chemistry–Climate Model Initiative (CCMI) predict the return of polar Antarctic total column ozone to 1980 values for the period of 2055 to 2066

(Dhomse et al., 2018).

The enhanced ozone depletion during SH spring is enabled by the formation of polar stratospheric clouds, acting as a surface for heterogeneous chemistry, activating halogens from ODSs that catalytically destroy ozone when the Sun comes back to the high latitudes in spring (Solomon et al., 1986). Ozone depletion positively feeds back on the anomalously low temperatures in the lower polar stratosphere by reducing the absorption of solar radiation in that region (e.g., Shine, 1986; Ramaswamy et al.,

1996; Randel and Wu, 1999), which in turn can lead to enhanced ozone depletion. In addition to ozone depletion, also the increase in greenhouse gas (GHG) concentrations contributes to low temperatures in the stratosphere (Fels et al., 1980). However, while ozone depletion and the connected radiative cooling are constrained mainly to the lower stratosphere, GHG–induced cooling spreads throughout the whole stratosphere. Both cooling effects can therefore have an influence on the dynamics of the stratosphere and possibly also on the troposphere. During the last decades of the $20^{th}$ century, along with the ozone depletion,

a positive trend in the Southern Annular Mode (SAM) was observed (Thompson and Solomon, 2002). This trend is connected to a strengthening and a poleward shift of the tropospheric jet (see reviews by, e.g., Thompson et al., 2011; Previdi and Polvani, 2014), that also affects the Southern Ocean (e.g., Sigmond and Fyfe, 2010; Ferreira et al., 2015). There have been a number of model studies aiming at separating the influence of GHGs and ODSs onto this observed trend of the tropospheric jet, i.e. the SAM (e.g., McLandress et al., 2011; Polvani et al., 2011b; Morgenstern et al., 2014; Solomon et al., 2017). McLandress et al.

(2011), for example, found that the observed SH ozone depletion had a significant impact onto the positive SAM trend during austral summer (December to February, DJF). Several studies agree that during this time of the year the impact from ODSs dominates over that from GHGs (e.g., McLandress et al., 2011; Polvani et al., 2011b; Solomon et al., 2017). Under ozone recovery conditions, that are projected for the upcoming decades, the radiative heating effects of ozone (positive) and GHGs (negative) will counteract each other (McLandress et al., 2011; Polvani et al., 2011a). However, when exactly ozone recovery is

strong enough to compensate GHG cooling is an open question and also depends on future GHG levels. Recent studies discuss the possibility that polar stratospheric ozone recovery started already (Solomon et al., 2016; Kuttippurath and Nair, 2017). The

recovery signal, however, is hard to detect and the impact of low ozone concentrations especially at polar southern latitudes will continue to influence atmospheric circulation in the near future (Bednarz et al., 2016).

A better understanding of the interaction between ozone chemistry and atmospheric dynamics is therefore crucial for future climate simulations. The way ozone is represented in climate models has a large impact onto the model's ability to simulate interactions between chemistry and dynamics. With this study we want to improve the knowledge about chemistry–climate interactions in the past to shed light onto how important the representation of ozone in climate models is also for future climate projections.

There are different ways to represent ozone in climate models: 1) Ozone can be calculated interactively using a chemistry scheme within a climate model. This is computationally very expensive, but the most appropriate representation of ozone and other trace gases, linking them directly with the radiation code and model dynamics. Models that implement such a chemistry scheme are referred to as chemistry–climate models (CCMs) and are commonly used for stratospheric applications such as in the WCRP–SPARC initiatives and the WMO ozone assessment reports. 2) Another way to represent ozone in a climate model is to prescribe it based on observed and/or modeled ozone fields, as provided, for example, by the IGAC/SPARC initiatives for the Climate Model Intercomparison Project, Phases 5 and 6 (CMIP5 and CMIP6; see Cionni et al., 2011; Checa-Garcia et al., 2018). As a consequence, the specified ozone field is normally not consistent with the internal model dynamics and does not allow for two–way interactions between ozone chemistry and atmospheric physics, since ozone is fixed and will not react to changes in transport, dynamics, radiation or temperature. Feedbacks between ozone concentrations and model physics are only possible if ozone is calculated interactively.

These feedbacks have been shown to contribute to shaping the response of atmospheric dynamics and modes of variability, such as the SAM, to SH ozone depletion by, for example, enabling the interaction between GHG cooling and ozone chemistry (Morgenstern et al., 2014). Others discuss the influence that chemical–dynamical feedbacks have on wave–mean flow interactions within the stratosphere (Manzini et al., 2003; Albers et al., 2013), including positive and negative feedbacks based on the strength of the background westerlies following the Charney–Drazin criterion (Charney and Drazin, 1961). Positive feedbacks can therefore only occur during strong westerly wind regimes. Under these conditions an additional cooling due to ozone depletion leads to a decrease in vertically propagating planetary waves, which further strengthens the polar vortex, further decreases the intrusion of ozone rich air masses from above and from lower latitudes and thereby further contributes to ozone depletion. Negative feedbacks come into play when the background westerlies are weak and an initial cooling due to ozone depletion would lead to an increase in upward wave propagation, decreasing the strength of the polar vortex and thereby increasing the intrusion of relatively ozone rich air masses. The negative feedback is especially important in spring (Manzini et al., 2003), since this is the time of the year when the westerly wind strength normally decreases and eventually turns easterly. Recently, such feedbacks have been discussed to be important also for surface climate variability on both hemispheres (Calvo et al., 2015; Lin et al., 2017; Haase and Matthes, 2019). Negative and positive feedbacks between chemistry and dynamics are discussed in detail in Haase and Matthes (2019) for the NH. They found especially the negative feedback at the end of the winter season to be important for the difference between specified and interactive chemistry simulations, which led to a more rapid and earlier stratospheric vortex break–down in the interactive chemistry simulations. Here, we will focus on the

sensitivity of SH climate and trends to the representation of ozone and the associated chemical–dynamical feedbacks.

In addition to the lack of feedbacks, prescribing ozone comes with other inaccuracies. Until recently it was recommended to use a zonally averaged, monthly mean ozone field as an input in ocean–atmosphere coupled climate models (CMIP5; see
Cionni et al., 2011). This neglects temporal and spatial variabilities in atmospheric ozone concentrations. Using monthly mean fields introduces biases in the model's ozone field that reduce the strength of the actual seasonal ozone cycle due to the interpolation of the prescribed ozone field to the model time step. To reduce these biases, a daily ozone forcing can be applied as demonstrated in Neely et al. (2014). Seviour et al. (2016) showed that using a daily ozone forcing does not only increase the effect of ozone depletion on the atmospheric response but that an impact is also found in the interior of the ocean. Furthermore,
ozone is not distributed zonally symmetric in the real atmosphere, therefore prescribing zonal mean ozone values inhibits the effect that an asymmetric ozone field can have onto the dynamics (Albers and Nathan, 2012). Different studies showed that including 3–dimensional (3D) ozone in a model simulation would lead to a cooler and stronger SH polar vortex during austral spring and/or summer (Crook et al., 2008; Gillett et al., 2009). The recommended ozone forcing for CMIP6 now uses a derived 3D ozone field, but does not include variability on time scales smaller than a month (Checa-Garcia et al., 2018).

Since a dynamically consistent representation of ozone that does not require an interactive chemistry scheme is of large interest to the scientific community, alternative methods of ozone representations are considered in the literature. For example, an online parameterization or simplified online scheme for ozone can be applied. This is a step in between a fully–interactive and a specified chemistry setup and allows the ozone field to follow the dynamics to a certain degree, e.g., as in CNRM–CM6 (Voldoire et al., 2019) or E3SM–1–0 (Golaz et al., 2019). Another possibility is described in Nowack et al. (2018), who apply
machine learning to achieve a higher consistency between the model's ozone field and the actual climate state of the model for specific scenarios. Also worth mentioning is Rae et al. (2019), who designed a computationally efficient method to interactively re–scale prescribed ozone values to a dynamically model–consistent 3D ozone field based on the potential vorticity field of the model. This method, unfortunately, is not well suited to represent the observed SH ozone depletion since it follows a solely dynamical approach and has therefore difficulties to account for heterogeneous chemistry processes. Therefore, until now a
fully–coupled chemistry scheme is the only way to guarantee for the complete range of chemical–dynamical interactions.

For the investigation of the SH ozone trend and its effect onto the tropospheric jet, different representations of ozone were applied in climate model studies. Recently, Son et al. (2018) compared different high–top CMIP5 models, and the latest CCMI model simulations with and without an interactive ocean, with regard to their representation of the tropospheric jet response to SH ozone depletion. They found that all models capture the poleward shift and intensification of the tropospheric jet in re-
sponse to ozone depletion. Nevertheless, Son et al. (2018) also point out that there is a large inter–model spread in the strength of the jet shift and intensification, partly due to differences in the ozone trends, but also influenced by differences in the model dynamics. The degree to which interactive versus specified chemistry plays a role for the tropospheric jet response to ozone depletion can not be inferred from such a multi–model study. In another multi–model study, Seviour et al. (2017) argue that interannual variability is very strong and large ensembles or long time slice simulations are required to detect robust differences
among models regarding the signal in the troposphere from stratospheric ozone depletion. Therefore, to assess this problem, we focus on a 9–member ensemble using a single CCM; the Community Earth System Model, version 1 (CESM1), with the

Whole Atmosphere Chemistry Climate Model (WACCM) as its atmosphere component. Using this model, Calvo et al. (2017) showed that reducing the SH cold pole bias in WACCM leads to a better representation of the ozone and accompanied temperature trends in the stratosphere. They attribute the improvement of the temperature trend to an increase in dynamical heating by a strengthened Brewer–Dobson–Circulation (BDC). The additional warming has two effects: 1) a direct effect onto the temperature reducing the cooling trend and 2) an indirect effect by reducing ozone depletion and therefore increasing radiative heating in spring. The second effect is due to interactions between chemistry and dynamics which would not be possible in a model without interactive chemistry.

However, studies that systematically assess the importance of interactive chemistry on the representation of tropospheric trends are very sparse. One of the first studies addressing this issue was carried out by Waugh et al. (2009). Using NASA's Goddard Earth Observing System Chemistry–Climate Model (GEOS CCM) to investigate the effect of SH ozone trends on the atmospheric circulation, they found a stronger cooling (warming) trend in the stratosphere for ozone depletion (recovery) with interactive chemistry and an underestimation of Antarctic temperature trends and trends in the SAM when ozone was prescribed as a monthly mean in the CCM. Li et al. (2016) confirmed the results from Waugh et al. (2009) coupling version 5 of the same CCM (GEOS–5) to an interactive ocean. They compared the interactive chemistry version of the model to a specified chemistry version of the same model, using monthly mean, zonal mean ozone values from the interactive chemistry simulation. Apart from ozone also other radiatively active species were prescribed in the specified chemistry version of the model. They found a statistically significant stronger cooling trend in austral summer in the lower stratosphere for the period of 1970 to 2010 when interactive chemistry was included in the model. This was accompanied by a stronger trend in the tropospheric jet stream strength, which increased towards the surface, also impacting the ocean circulation. They argue that the stronger lower stratospheric temperature trend was due to a stronger negative ozone trend in the interactive chemistry simulation resulting from either using a monthly mean ozone field (Neely et al., 2014) and/or from excluding asymmetries in the ozone forcing (e.g., Crook et al., 2008; Gillett et al., 2009). The weaker tropospheric trends in the specified chemistry model version were therefore partly due to a weaker ozone forcing compared to the one in the interactive chemistry version. To isolate the effects that ozone feedbacks have, a different experimental setup is required.

Here, we use an interactive chemistry climate model and its specified chemistry counterpart with a transient zonal mean daily ozone forcing to investigate the effects of interactive chemistry onto the stratospheric and tropospheric temperature and zonal wind trends due to ozone depletion. We use a daily ozone forcing to reduce the difference of the ozone forcing between the specified and interactive chemistry simulations. Additionally, a sensitivity experiment using a transient daily 3D ozone field in the specified chemistry version is applied to assess the impact that ozone asymmetries have in this experimental setting. An ensemble of 9 members for each experiment is used to better capture the forced response.

The paper is organized as follows: Section 2 introduces the model simulations and methods applied in this study. The impacts of interactive chemistry and chemical–dynamical feedbacks onto the climatology and trends due to SH ozone depletion are analysed in section 3. Additionally, the sensitivity of the tropospheric jet response to ozone depletion under different chemistry settings (daily zonal mean vs. daily 3D ozone) is investigated. We conclude our findings with a summary and discussion in section 4.

## 2 Data and Methods

Similar to Haase and Matthes (2019), we use NCAR's CESM1 model, with WACCM version 4 as the atmosphere component (CESM1(WACCM); Marsh et al., 2013)). CESM1(WACCM) is a fully coupled climate model with interactive ocean, land and
sea ice components. For a detailed description of the model setup we refer to Haase and Matthes (2019) and references therein. WACCM4 is a fully–interactive CCM, which reproduces stratospheric dynamics and chemistry very well (Marsh et al., 2013). Nevertheless, WACCM4 has, like many other CCMs, a cold pole bias on the SH, which leads to a stronger and longer lasting polar vortex as compared to observations on the SH (Richter et al., 2010). This bias also influences the strength of the simulated ozone hole since ozone depletion can be more effective/severe under lower temperature conditions. At the same time, mixing
of ozone rich air masses into the polar regions is inhibited by a strong polar night jet (PNJ), reducing ozone concentrations further.

Therefore, in this study, an improved version of WACCM4 was used. We implemented a few modifications in the model code published in Garcia et al. (2014); Smith et al. (2015) and Garcia et al. (2017): 1) the dependency of the orographic gravity wave drag on land fraction was removed at all latitudes; 2) the Prandtl number was increased, which increases diffusion and
thereby influences the downward transport of trace gases at the winter pole; and 3) the portion of energy from gravity wave dissipation, that is transformed into heat was reduced from 100% to 30%. These improvements help to reduce the cold pole bias in the model upper stratosphere by 2.5 K in the annual mean in a pre–industrial control setting (Suppl. Fig. 1). Our version of WACCM does not include all modifications introduced by Garcia et al. (2017). Namely, it still lacks the impact of the updated chemistry scheme and does not include all of the adjustments made to the gravity wave parameterizations (only those
mentioned above, since these were known to us when the experiments were performed). Therefore, this model version is not the same as the so–called WACCM-CCMI version described in Calvo et al. (2017), but a step in between the CMIP5 version of WACCM (WACCM4) and WACCM-CCMI. Despite the remaining differences to WACCM-CCMI (see Supplement), the reduction of the cold pole bias (by 2.5 K in the annual mean) and the weakening of the PNJ, by about 9 $\mathrm{ms}^{-1}$ in the annual mean, is significant (Suppl. Fig. 1). The impact of the model adjustments to the seasonal mean zonal mean temperature and
zonal mean zonal wind climatologies can also be found in the Supplement (Suppl. Fig. 2).

Apart from theses adaptations, WACCM4 is used in its standard configuration at a horizontal resolution of 1.9°latitude by 2.5°longitude and 66 levels in the vertical up to the lower thermosphere (upper lid at $5.1\mathrm{x}10^{-6}$hPa or about 140 km) as described in Haase and Matthes (2019). The chemistry in this configuration is still based on the Model for Ozone and Related Chemical Tracers, version 3, (MOZART3; Kinnison et al., 2007). The Quasi-Biennial Oscillation (QBO) is not generated
internally, and hence in our simulations stratospheric equatorial winds were relaxed towards an idealized QBO with a fixed periodicity of 28 months as described in Matthes et al. (2010).

### 2.1 Model Simulations

To investigate the importance of interactive chemistry on the impact of ozone depletion on the SH jet, we performed three sets of experiments as summarized in Table 1. The first set used the interactive chemistry version of CESM1(WACCM) as described

in the previous section, while the other two sets used the specified chemistry version of WACCM (SC-WACCM, Smith et al., 2014). All simulations were performed in a fully–coupled setup with the same interactive ocean, land and sea ice components. In SC-WACCM, the interactive chemistry scheme is turned off and feedbacks between chemistry and model physics are not represented. The improvements implemented in WACCM (as described above) were also used in our SC-WACCM simulations. All other settings are equal to those applied in Haase and Matthes (2019) and are therefore not addressed in detail again. But we would like to mention that the ozone concentrations for the whole atmosphere and concentrations of other radiatively active species as well as the total short–wave heating rates above 65 km that are prescribed in the SC-WACCM simulations (Smith et al., 2014), are derived from the interactive chemistry WACCM simulations used in this study. For ozone, transient daily–resolved ozone mixing ratios are prescribed throughout the whole atmosphere. We will refer to the interactive chemistry version of CESM1(WACCM) as "Chem ON" and to the specified chemistry version, that uses SC-WACCM as the atmosphere component, as "Chem OFF". To account for the impact of asymmetries in ozone, we also include a set of sensitivity experiments where we prescribe a 3D transient daily ozone field in SC-WACCM. This experiment is referred to as Chem OFF 3D. Apart from using 3D ozone instead of a zonal mean ozone field, all other settings are equal between Chem OFF and Chem OFF 3D. In contrast to Haase and Matthes (2019), we ran a total of 9 ensemble members for each experiment to improve the significance of the presented results. The specified chemistry setup runs about 4 times faster than the full chemistry setup and is therefore computationally much cheaper.

As the focus of this study is on the impact of observed lower stratospheric ozone trends onto the circumpolar jet in the SH, our experiments are carried out based on historical forcing conditions for 1955 to 2005 and on the representative concentration pathway 8.5 (RCP8.5) for the period of 2006 to 2013. Hence, the simulations cover a 58–year period that covers the period in which catalytic ozone depletion started and before ozone recovery becomes important. The external forcings are mostly based on the CMIP5 recommendations: GHG and ODS concentrations (Meinshausen et al., 2011), as well as volcanic aerosol concentrations (Tilmes et al., 2009). However, for the spectral solar irradiances and the geomagnetic activity as proxy forcing for energetic particle effects the more recently published CMIP6 forcing was applied (Matthes et al., 2017).

## 2.2 Observational Data

To verify our modeled temperature trend, we compare it with observational temperature data from the Integrated Global Radiosonde Archive, version 1 (IGRA) from the National Centers for Environmental Information (NCEI) of the National Oceanic and Atmospheric Administration (NOAA). The earliest data records in IGRA go back to 1905. However, time records as well as the temporal and vertical resolution differs between the stations included in this archive (Durre et al., 2006). The IGRA data used in this study covers 17 different height levels from the surface up to 10 hPa and only a selected time period from 1969 to 1998 is considered. It has to be noted that the spatial distribution of the IGRA stations is rather sparse in the SH, especially over higher latitudes. However, there is a good agreement of the maximum negative temperature trend between the IGRA data and estimates from other radiosonde products Young et al. (2013, see Table 2).

## 2.3 Methods

Our analysis focuses on the evaluation of climatologies and linear trends, in particular for the SH ozone trend and its impact on other climate variables. The climatologies and trends for the different experiments are calculated from the ensemble average of all nine ensemble members.

Climatological differences between the simulations with and without interactive chemistry are displayed as: Chem ON minus Chem OFF for the time period 1955 - 2013 to illustrate the effects of interactive chemistry. Statistical significance at the 95% level is tested using a two-sample t-test. Furthermore, we consider lead–lag correlations between ozone at 50 hPa and polar cap dynamical heating rates at each level for the same time period for each ensemble member separately. Before the correlation coefficients are calculated a slowly-varying climatology (Gerber et al., 2010) is removed from the data to avoid correlating trends. Afterwards, the ensemble mean of the correlation coefficients is calculated. Statistical significance is chosen to be given for each point in which at least 5 out of 9 individual ensemble members reach a p-value $\leq 0.05$.

The SH trends for polar cap temperature, heating rates, and zonal mean zonal wind ($60°$- $70°$S) are calculated for the period of 1969 - 1998, which is marked by a strong ozone decline in October in the SH lower polar stratosphere (Fig. 4a). We restrict the trend analysis to this period for a better comparison to earlier model and observational studies (e.g., Calvo et al., 2012; Young et al., 2013; Calvo et al., 2017). To determine the statistical significance of the linear trend differences, a new time series is produced by taking the difference between the time series of the ensemble means. This approach reduces noise levels by subtracting the variability of the individual time series and favors the identification of real trend differences (Santer et al., 2000). The trend significance is estimated using the commonly used Mann–Kendall test at a confidence level of 95%.

For the tropospheric jet trend, we use jet latitude and strength at 850 hPa, which were calculated applying a quadratic fit to the maximum grid point and the two adjacent points either side following the procedure described, e.g., in Simpson and Polvani (2016).

To address the impact of interactive chemistry on inter–annual variability, the timescale of the Southern Annular Mode (SAM) is evaluated for Chem ON, Chem OFF, and Chem OFF 3D following the procedure of Simpson et al. (2011) and Ivanciu et al. (2020). The SAM index used for this calculation is determined for each ensemble member separately and follows the definition by Gerber et al. (2010) using the first EOF of daily zonal mean geopotential height, which is previously adjusted by removing the global mean and a slowly varying climatology to remove variability on decadal timescales. For the calculation of the SAM timescale, the autocorrelation function of each SAM index is calculated and smoothed. Then the e-folding timescale is estimated by using a least squares fit to an exponential curve up to a lag of 50 days to the smoothed autocorrelation function (Simpson et al., 2011; Ivanciu et al., 2020).

## 3 The impact of stratospheric chemistry on southern hemispheric climate and trends

Haase and Matthes (2019) (in the following referred to as HM19) showed that including interactive chemistry leads to a stronger and a colder polar stratospheric vortex on both hemispheres. The differences between the interactive and specified chemistry simulations were shown to be largest during mid-winter and in spring when ozone chemistry gets important. These

results were based on only one model realization per experiment. Here, an ensemble of 9 realizations per experiment is used to evaluate the impact of interactive chemistry on the SH climatology and trend. In a first step the climatological difference between Chem ON and Chem OFF is analysed for the whole model period (1955-2013). Figure 1 shows the seasonal evolution of zonal mean zonal wind at 10 hPa and zonal mean temperature at 30 hPa similar to Figure 2 in HM19. It shows that the main results presented in HM19 are reproduced by the 9–member ensemble. Including interactive chemistry leads to a stronger PNJ (Fig. 1a) and a colder polar stratospheric vortex (Fig. 1b) on both hemispheres. The significance of this difference is larger as compared to HM19, while the amplitudes of the differences are smaller. This is not an unexpected feature from taking the average over 9 ensemble members compared to only considering a single realization since averaging reduces the imprints of natural variability; the forced signal is therefore easier to detect. In the Chem ON ensemble, a significantly stronger PNJ is apparent from September until April in the NH, and from September to December in the SH (Fig. 1a). The months that show the largest differences between the interactive and specified chemistry ensemble agree well with HM19: January and March in the NH and October to November in the SH. The impact of interactive chemistry on lower stratospheric temperatures is even more significant showing a cooler polar lower stratosphere covering almost the whole year (with the exception of January and February in the SH) and a warmer lower stratosphere between $40°S$ and $40°N$ (Fig. 1b). This result is consistent with a weaker shallow branch of the BDC in the model experiment with interactive chemistry discussed in HM19 (see also Suppl. Fig. 3).

## 3.1   Stratospheric mean state

The climatological differences for the SH polar stratosphere are depicted in Figure 2. Although the ozone and short–wave (SW) heating climatologies are almost identical between Chem ON and Chem OFF (not shown), there is still a difference in the climatology of the polar cap temperatures that is also imprinted in the strength of the circumpolar jet (Fig. 2a and b). The temperature difference is characterized by lower values in the lower and middle stratosphere from May until November, with maximum differences in September and October, followed by higher values peaking in December (Fig 2b). This pattern compares well with the one found by Neely et al. (2014) but shows a higher statistical significance, also covering the stratospheric levels above 30 hPa during all seasons. As mentioned earlier, this temperature difference is also reflected in the strength of the circumpolar jet (Fig. 2a), which is stronger in Chem ON, especially during November and December when the strength of the polar vortex normally starts to decrease. Following HM19, long–wave (LW) and dynamical heating rate climatologies are considered to investigate the polar cap temperature difference between Chem ON and Chem OFF (Fig. 2c and d) in more detail. In agreement with the findings of HM19 for the NH, Figure 2d shows that also on the SH, the dynamical heating rates are responsible for the temperature differences between Chem ON and Chem OFF, whereas the LW heating rates tend to damp the temperature tendencies caused by the dynamics (Fig. 2c).

The impact of the dynamics onto the mean state of the stratosphere suggests that similar feedbacks as compared to the NH can be expected also for the SH. Figure 3 shows a lag correlation between polar cap ozone at 50 hPa and the dynamical heating rates with ozone leading the dynamics by 15 days following the procedure in HM19. The climatological zonal mean zonal wind for values $\leq 20 \text{ ms}^{-1}$ is also depicted (contours). The negative correlation between ozone and dynamical heating rates represents the negative feedback discussed earlier: under weaker westerly wind background conditions, ozone depletion and

the associated radiative cooling lead to a westerly acceleration in the lower stratosphere that enhances upward wave propagation and dissipation which eventually leads to an earlier break–down of the stratospheric polar vortex. This feedback is also apparent in the Chem OFF simulation but it is weaker in amplitude. This is different compared to the findings for the NH, where the negative feedback was not found for the specified chemistry version of the model. We suppose that the presence of this correlation in Chem OFF is due to the fact that a part of the negative feedback is included in the prescribed ozone field, which is characterized by a strong negative trend in ozone (see following section), which dominates ozone variability on the SH. Apart from the negative correlation, also a positive correlation during stronger westerly background winds can be detected in Figure 3a in the lowermost stratosphere. It is less significant than the negative correlation but could be regarded as a hint for the positive feedback between ozone and the dynamical heating rates in Chem ON.

Figures 1 and 2 showed that including interactive chemistry leads to a stronger circumpolar jet and a colder polar stratospheric vortex, especially towards the end of the vortex lifetime. The differences between Chem ON and Chem OFF are mainly due to differences in dynamical heating, which we attribute at least partly to the representation of chemical–dynamical interactions (feedbacks). Especially, the dominant negative feedback, which starts in November in the upper stratosphere and peaks in January in the lower stratosphere is stronger in Chem ON contributing to the enhanced dynamical heating in this ensemble, which also starts in November (Fig. 2). Since this period is strongly influenced by ozone depletion (see Suppl. Fig. 4 for a climatology of the pre–ozone hole period), we also expect an impact of chemical–dynamical interactions onto the stratospheric and tropospheric trends associated with ozone depletion. This will be the focus for the remainder of our analysis.

## 3.2 Stratospheric trends

Figure 4a exemplarily depicts the temporal evolution of ozone mixing ratios at 50 hPa in October, which represents the maximum ozone depletion in our model simulations (Fig. 4b). The ozone trend in Chem ON agrees well among the different ensemble members (gray lines in Fig. 4a), starting of with a weak negative trend from 1955 to the late 1960s, followed by a strong negative trend, which levels off in the mid-1990s. To address the model's response to SH ozone depletion the period of 1969 to 1998 is chosen (red line in Fig. 4a), as it covers the period of strongest ozone depletion. This period is also chosen to facilitate comparisons to earlier studies using the WACCM model or observational data (Table 2).

Due to the ozone depletion from 1969-1998, which reaches its maximum of about -0.9 $\mathrm{ppmv\,decade^{-1}}$ in the lower stratosphere during October (Fig. 4b) a decrease in polar lower stratospheric temperatures can be observed (Fig. 5). In Chem ON the negative temperature trend maximizes with -6.6 $\mathrm{K\,decade^{-1}}$ in December at about 90 hPa (Fig. 5a) and is therefore stronger and delayed by one month compared to observations, which show a maximum trend of -4.0 $\mathrm{K\,decade^{-1}}$ during November at about 100 hPa (IGRA; Fig. 5b and Table 2). The overestimated temperature trend in Chem ON, however, is quite common in CCMs (e.g., Eyring et al., 2010; Young et al., 2013). It compares well to the published WACCM4 trend (see Calvo et al., 2012, 2017, and Table 2) but is larger than the trend found in WACCM-CCMI (Calvo et al., 2017). The reduction of the trend from WACCM4 to WACCM-CCMI can be explained by a reduction of the cold pole bias in the model (Calvo et al., 2017). Although a reduction of the cold pole bias was also achieved in our WACCM version by implementing a few changes to the model code (see Methods and Supplement for details), the trend is not significantly weaker compared to the original WACCM4

version analyzed in Calvo et al. (2012). In agreement with the overestimated temperature trend, also the ozone trend is with -0.9 ppmv decade$^{-1}$ (Fig. 4b) rather in agreement with WACCM4 than with WACCM-CCMI (Calvo et al., 2017). This indicates that the reduction of the cold pole bias implemented here, is not sufficient to reproduce the WACCM-CCMI trend. However, the comparison between our ensembles of Chem ON and Chem OFF simulations is still very suitable to address the question of how important ozone feedbacks are for the stratospheric and tropospheric circulation.

The negative temperature trend due to ozone depletion is followed by a positive temperature trend at altitudes above 30 hPa in the model and observational data (Fig. 5). This positive temperature trend coincides with a positive ozone trend (Fig. 4b). The ozone trend, however is not the only contributor to this temperature trend pattern. Figure 6 depicts the different heating rate trends, that combine to the temperature trend pattern. The SW heating rate trend (Fig. 6a) resembles the trend in ozone (Fig. 4b) during the time of the year when solar radiation is available at such high latitudes. It explains the negative temperature trend in the lower stratosphere and parts of the positive temperature trend following it in the upper stratosphere. However, also long–wave (LW, Fig. 6b) and dynamical (DYN, Fig. 6c) heating rate trends contribute to the temperature trend. Especially, the dynamical heating rate trend is decisive for that part of the temperature trend pattern that can not be explained by the SW heating trend. There is a strong positive trend in dynamical heating starting in November in the upper stratosphere and propagating down to about 100 hPa in January. This positive trend can be explained by a stronger descent of air masses through an increase in the residual meridional circulation, i.e. a strengthening of the BDC, during the ozone depletion period (e.g., Keeble et al., 2014). This dynamical response is due to the negative feedback (compare Fig. 3) that evolves due to the extension of the stratospheric vortex lifetime and the connected wave forcing (Manzini et al., 2003; Oman et al., 2009; Albers and Nathan, 2013; Lin et al., 2017; Haase and Matthes, 2019). A significant negative trend in the dynamical heating in the lowermost stratosphere during November and December is indicative of a positive feedback between ozone chemistry and the model dynamics (Lin et al., 2017), which is in agreement with the positive correlation in Figure 3a. The LW heating rate trend mostly damps the signals from the SW and dynamical heating rate trends (Fig. 6).

Is this feedback loop at all represented in Chem OFF? Figures 7a and b show the 1969-1998 temperature trend for Chem OFF and the difference of the trend between Chem ON and Chem OFF. By construction, the polar cap ozone trend is the same between the two ensembles; and so is the trend in SW heating rates (not shown). Nevertheless, with a maximum of -5.9 K decade$^{-1}$ in November, the maximum temperature trend in Chem OFF is weaker compared to Chem ON (-6.6 K decade$^{-1}$) and occurs earlier, which could be due to the lack of a positive feedback when ozone is prescribed rather than calculated interactively (compare Fig. 3). This gets clearer when the difference between Chem ON and Chem OFF is considered (Fig. 7b): The largest differences occur in December and January, and are characterized by a longer lasting cooling trend in Chem ON in the lower stratosphere (positive feedback) as well as by a stronger warming trend starting in December in the upper stratosphere reaching down into the lowermost stratosphere in February and March (negative feedback). These differences can mainly be attributed to stronger trends in the dynamical heating rates in Chem ON (Figs. 7c and d), which is due to the better representation of feedbacks between chemistry and dynamics in the fully–coupled chemistry model setup.

We therefore conclude that, in accordance to the findings of HM19, the stronger dynamical warming in Chem ON can be explained by negative feedbacks between ozone chemistry and model dynamics. During weak westerly winds, an unusually low

ozone concentration can lead to enhanced upward planetary wave propagation by extending the lifetime of the westerly wind regime, which enhances a descent over polar latitudes resulting in an additional adiabatic warming. Apart from the negative feedback, which was found to be apparent also in the NH, a positive feedback can be detected in the SH ozone depletion period during stronger westerly background winds. It is statistically significant only for 7 out of 9 members (not shown) and restricted to the lowermost stratosphere. This positive correlation is only found in the interactive chemistry setup and could explain the stronger dynamical cooling in Chem ON compared to Chem OFF (Fig. 7). It has to be noted that Chem OFF is able to represent the negative feedback pattern to a certain extent (Fig. 3) because of the strong ozone signal that the model is forced with (parts of the feedback can be considered to be included in the prescribed ozone fields).

The negative temperature trend in the lower polar stratosphere increases the meridional temperature gradient and leads to a strengthening of the PNJ, especially towards the end of the polar vortex lifetime as discussed before. Figures 8a and b show that the maximum trend in zonal mean zonal wind in the PNJ region is stronger in Chem ON ($9.2 \, \mathrm{ms^{-1} \, decade^{-1}}$) compared to Chem OFF ($7.8 \, \mathrm{ms^{-1} \, decade^{-1}}$). The largest differences in the zonal mean zonal wind trend can be found in the middle stratosphere during December (Fig. 8c), which supports our earlier argumentation about the characteristics of chemical–dynamical feedbacks in the two ensembles. Namely, that an extension of the vortex lifetime, which is stronger in Chem ON compared to Chem OFF, favors the occurrence of the negative feedback. However, in both ensembles a significant zonal mean zonal wind trend can be found also at the surface from November through February, which will be investigated in the following with a focus on the austral summer season (DJF).

### 3.3 Tropospheric jet trend and SAM timescale

Figure 9 shows the 1969-1998 trend for zonal mean zonal wind with latitude and height (color shading) along with the climatological wind over the same period (contours) for DJF. It is evident that the strengthening of the PNJ is connected also to a strengthening of the tropospheric jet and its poleward displacement in agreement with earlier studies (e.g., Thompson and Solomon, 2002; Son et al., 2008; Eyring et al., 2013). In Chem OFF, the strengthening of the poleward flank of the tropospheric jet compares well to the signal in Chem ON, but the weakening of the equatorward flank is weaker. Hence, part of the differences found in the stratosphere also have an effect onto the troposphere.

Apart from the insufficient representation of chemical–dynamical feedbacks in the Chem OFF ensemble, also the decision to prescribe zonal mean ozone can have an impact onto the characteristics of the tropospheric jet trend associated with ozone depletion (Crook et al., 2008; Gillett et al., 2009; Rae et al., 2019). We therefore, additionally consider an experiment that uses 3D ozone in the prescribed ozone fields (Chem OFF 3D, Figs. 9d and e). Including 3D ozone improves the representation of the circumpolar jet trend in response to ozone depletion in comparison to using a zonal mean ozone field. The strengthening of the poleward flank of the circumpolar jet is very well captured in the troposphere and stratosphere, while the weakening of the equatorward flank of the tropospheric jet is lower compared to Chem ON but much better represented compared to Chem OFF. Whether the difference in the mid–latitude DJF zonal mean zonal wind trend really impacts the trend of the tropospheric jet, is addressed in the following. Figure 10 shows the trend for the tropospheric jet latitude and strength at $850 \, \mathrm{hPa}$. There is no statistically significant difference between the chemistry settings in the trend of the tropospheric jet position and strength. All

experiments have a similar mean jet latitude trend and there is a large spread among ensemble members in the trend of the jet strength, which leads to hardly significant trends in the ensemble means. Therefore, the impact of interactive chemistry that is significant in the stratosphere does not seem to show the same significance in the troposphere.

Since the shift of the tropospheric jet is also manifested in a positive trend of the SAM (Thompson and Solomon, 2002), we use the SAM to investigate the connection between the stratospheric and tropospheric circulation from a different angle in the following. The SAM timescale (based on a detrended SAM index) is used to evaluate the impact of interactive chemistry on stratosphere–troposphere–coupling. It gives information about how persistent a SAM anomaly is in the atmosphere. Dennison et al. (2015), for example, showed that under ozone depletion the SAM timescale is enhanced and stratosphere–troposphere–
coupling is strengthened. Figure 11 shows the SAM timescale for the Chem ON, Chem OFF and Chem OFF 3D ensembles. There is a large difference in the stratospheric SAM timescale between these ensembles: Chem ON shows the largest persistence in the SAM, while Chem OFF shows the smallest. The notable reduction in the persistence of the stratospheric SAM in Chem OFF compared to Chem ON implies that feedbacks between chemistry and dynamics are of importance for this feature of the SAM. However, it has to be noted that the variability of the SAM timescale is large among the individual ensemble
members (see Suppl. Fig. 5). As expected from the previous results, the SAM timescale of the Chem OFF 3D ensemble is in between the other two ensembles. It represents the observations best (e.g., see Figure 2a in Simpson et al., 2011). Using ERA-Interim reanalysis data Simpson et al. (2011) show that the stratospheric SAM timescale maximizes with more than 72 days at around 50 hPa in October and peaks at the surface with a timescale of 14 days by the end of November. Chem ON overestimates the persistence of the SAM in the stratosphere with a timescale of more than 80 days from August to Novem-
ber, which is a common bias in CCMs (Gerber et al., 2010). But unlike most CCMs, Chem ON slightly underestimates the persistence of the SAM in the troposphere. This is in agreement with the findings of Gerber et al. (2010), who concluded that WACCM was one of the two models, that represented the tropospheric SAM best since all other CCMs in their study overestimated the impact of the stratospheric SAM onto the troposphere in the SH. However, the too short tropospheric SAM timescale in WACCM, which is found in all our experiments independent of the chemistry setting (Fig. 11), indicates that the coupling
between the stratosphere and troposphere is very likely too weak. This could explain why we do not find significant differences in the tropospheric jet trends between our experiments (Fig. 10 ) and that the largest impacts of the chemistry setting are found in the stratosphere.

  The SAM timescale is also used in Ivanciu et al. (2020) to evaluate the effects of interactive chemistry on the SAM. They compare a model with an interactive chemistry scheme to the same model using the monthly 3D CMIP6 ozone forcing. A different
experimental setup and model are used compared to our study. They investigate the impact of feedbacks between chemistry and dynamics as well as the issue of prescribing an ozone field (and trend) that is not consistent with the model dynamics. Similar to our results, Ivanciu et al. (2020) conclude that the SAM timescale is reduced in their Chem OFF ensemble compared to their Chem ON ensemble. But in their case, this signal also reaches down to the troposphere. These results support our conclusion that feedbacks between chemistry and dynamics as well as asymmetries in ozone have a significant impact on the
SAM timescale, but also imply that CESM1(WACCM) might not to be suited to investigate the effect of interactive chemistry on the tropospheric jet in the SH.

## 3.4 Effect of ozone asymmetries

To better understand the improvement in the Chem OFF 3D ensemble over the Chem OFF ensemble we consider spatially asymmetric trends of temperature, SW, LW and dynamical heating rates in the following. We focus on the region showing the largest differences between Chem ON and Chem OFF: the lower stratosphere (at 50 hPa) during December (Fig. 7d). Figure 12 shows that the negative temperature trend at 50 hPa is not entirely zonally symmetric. It is characterized by a zonal wavenumber 1 (wave–1) anomaly with a stronger cooling towards South America (over the Antarctic Peninsula) than towards Australia. A wave–1 pattern in the lower stratospheric temperature trend was also described in Lin et al. (2009). They found that ozone cooling and dynamical warming were affecting different locations around Antarctica.

The wave–1 pattern is also visible in the difference between Chem ON and Chem OFF (Fig. 12). Chem OFF 3D much better resembles the departure of the maximum cooling region towards the Antarctic Peninsula than Chem OFF. We attribute the deficiency to reproduce this wave–1 pattern in Chem OFF to the fact that only a zonal mean ozone field is prescribed to the model. This leads to differences in the SW heating trend between Chem ON and Chem OFF that is not apparent between Chem ON and Chem OFF 3D (Fig. 12). We suspect that the spatially asymmetric SW heating that is missing in Chem OFF leads to the apparent difference in the dynamical heating rate trend, which is leading to a stronger cooling over the Antarctic Peninsula in Chem ON (by up to -1.44 K $decade^{-1}$). The LW heating rate trend (Fig. 12) dampens the signal from the dynamical heating rate trend. The weaker dynamical heating in the lower stratosphere in Chem ON (compare Fig. 7d) was previously discussed to be part of the positive chemical–dynamical feedback (Fig. 3), that is possibly responsible for the stronger lower stratospheric temperature trend when interactive chemistry is included. The spatially asymmetric trends indicate that zonal asymmetries in SW heating additionally contribute to this feedback.

To summarize, stratospheric trends of polar cap temperature and zonal mean zonal wind are influenced by chemical–dynamical feedbacks in such a way that including these feedbacks (Chem ON) leads to a stronger cooling in the lower stratosphere in December (positive feedback) and to a stronger warming above (negative feedback) reaching into the lower stratosphere in January, which leads to a longer lasting (more persistent) polar vortex during the ozone depletion period when interactive chemistry is included. Apart from chemical–dynamical feedbacks, spatial asymmetries in ozone also play a role in shaping the atmospheric dynamical response to ozone depletion. Prescribing a 3D ozone field instead of a zonal mean field substantially improves the response of the circumpolar jet to ozone depletion. In accordance with Calvo et al. (2017), we find the dynamical response to the ozone depletion to be of particular importance for the stronger trend signals in Chem ON.

## 4 Conclusions

We investigated the sensitivity of the Southern hemisphere (SH) circumpolar jet response to ozone depletion under different representations of ozone chemistry in a climate model. For this purpose we used NCAR's CESM1(WACCM), a state–of–the–art coupled chemistry–climate model in its standard version including interactive chemistry (Chem ON) and in its specified chemistry version that uses a prescribed ozone field instead (Chem OFF). We ran a CCM ensemble of 9 members per experiment, in order to be able to detect the ozone depletion signal from internal variability. By prescribing daily ozone in the

specified chemistry version of WACCM instead of monthly mean values we reduce the difference in ozone forcings between the Chem ON and Chem OFF ensemble that would otherwise occur through linear interpolation to the model time step (Neely et al., 2014). Such an interpolation can lead to a reduction of the ozone hole strength and therefore also to a reduction of the stratospheric temperature trend due to larger short–wave (SW) heating rates. Such a causality was described in Li et al. (2016). In our setup, the SW heating rate trend due to ozone depletion in the period from 1969 to 1998 is almost identical between Chem ON and Chem OFF. Nevertheless, we still find a stronger cooling trend in the lower stratosphere when interactive chemistry is included. This also feeds back onto the circumpolar jet during this period. We attribute this difference to the better representation of chemical–dynamical feedbacks in Chem ON, which result in a longer lasting polar stratospheric vortex. Similar as in Haase and Matthes (2019), positive feedbacks as well as negative feedbacks are suggested to be of relevance. During December, lower temperatures in the lower stratosphere in Chem ON are due to a weaker dynamical heating, which can be attributed to a positive feedback mechanisms, whereas higher temperatures in Chem ON, which start in December in the middle stratosphere and reach the lower stratosphere in January, can be attributed to negative feedbacks between chemistry and dynamics. Apart from the differences in the long–term trend, also the inter–annual variability is affected by feedbacks. The persistence of the stratospheric SAM is significantly larger when interactive chemistry is included, in agreement with Ivanciu et al. (2020).

A sensitivity simulation with a prescribed daily 3D ozone field (Chem OFF 3D) was used to assess the importance of spatial asymmetry effects. It was found that, the stronger temperature trend in Chem ON is connected partly to the wave–1 structure of the SW heating rate trend due to ozone depletion. However, the asymmetric ozone structure does not explain all of the differences found between Chem ON and Chem OFF, which highlights the importance of feedbacks between chemistry and dynamics.

Our findings support the results by Li et al. (2016) that part of the stronger stratospheric temperature trend with interactive chemistry is due to missing asymmetries in the zonal mean ozone forcing. However, Li et al. (2016) used a monthly mean ozone forcing that led to a deeper ozone hole and larger SW heating trend in their interactive chemistry simulation. The differences they described for interactive versus specified chemistry were influenced by the differences in the ozone field. Using a daily ozone forcing we reduced the difference in the SW heating rate trend substantially between Chem ON and Chem OFF and still find a significantly stronger circumpolar jet response to ozone depletion in the interactive chemistry simulation. This shows that feedbacks between chemistry and dynamics are important for the characteristics of the SH circumpolar jet trend and should be considered when estimating the atmospheric response to future ozone recovery.

However, the impact of interactive chemistry on the tropospheric jet could not be validated by our study. This might be due to the weak stratosphere–troposphere–coupling in the model that is indicated by the low tropospheric time scale of the SAM. This feature might be connected to the interactive ocean, which shows large biases in sea ice retreat in the seasonal cycle (Landrum et al., 2012; Marsh et al., 2013). However, a recent study by Gillett et al. (2019) showed that the response between ozone depletion and the SAM was independent from coupling an interactive ocean to WACCM or running it with observed SSTs. They found an improvement in SAM teleconnections, though, in an updated version of the atmosphere model, namely CAM5. This implies that biases in the atmospheric physics might be responsible for the missing link to the tropospheric jet in

our study.

Although not directly affecting the position of the tropospheric jet, the differences we find between the chemistry settings (Fig. 9 ), show a stronger tropospheric response to ozone depletion when interactive chemistry is included. An updated model version of WACCM, based on the CAM5 physics, might improve our understanding of the stratospheric impact onto the troposphere under different chemistry settings.

*Code and data availability.*  The IGRA radiosonde data used in this paper is publicly available at https://www1.ncdc.noaa.gov/pub/data/igra/v1/. Pre-processed model data to reproduce the figures in the manuscript can be found under https://doi.org/10.5281/zenodo.3785404. Further CESM1(WACCM) model data requests should be addressed to Katja Matthes (kmatthes@geomar.de). The scientific code will be shared upon request to Sabine Haase (shaase@geomar.de).

*Author contributions.*  SH wrote the manuscript. KM and SH decided about the analysis and experimental design. TK and SW carried out
the model simulations. JF carried out the data analysis and produced all the figures. All co-authors commented on the manuscript.

*Competing interests.*  The authors declare that they have no competing interests.

*Acknowledgements.*  We thank Ioana Ivanciu for helpful discussions especially on the SAM timescale. We thank the computing center at Christian–Albrechts–University in Kiel and at the Deutsches Klimarechenzentrum (DKRZ) for support and computer time.

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

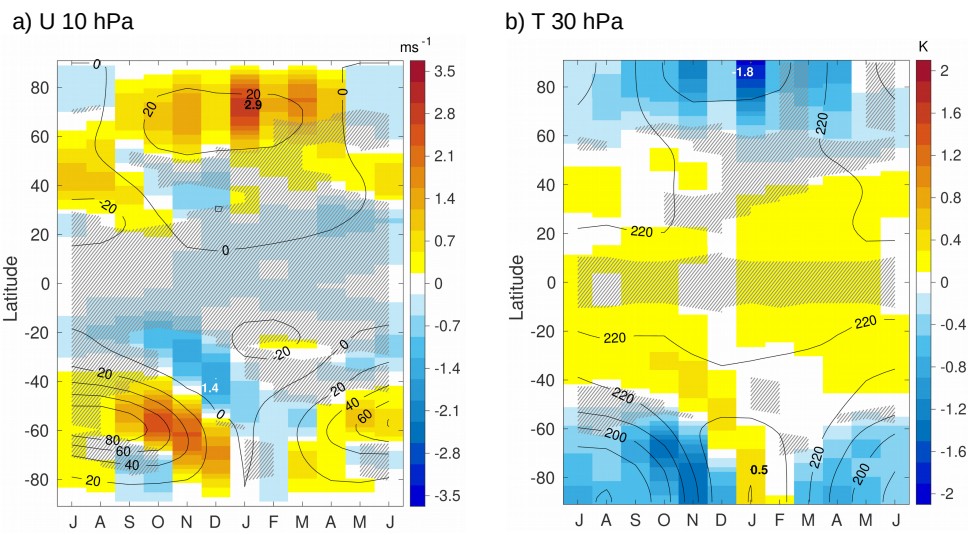

**Figure 1.** Monthly ensemble mean differences for 1955 to 2013 between Chem ON and Chem OFF for the climatological zonal mean zonal wind (U) at 10 hPa in $\mathrm{ms^{-1}}$ (a) and zonal mean temperature (T) at 30 hPa in K (b) as a function of latitude and month (shading). Contours represent the climatological mean state for Chem ON. The contour intervals are $20\,\mathrm{ms^{-1}}$ (a) and 20 K (b). Statistically insignificant regions are hatched at the 5% level based on a two-sample t-test.

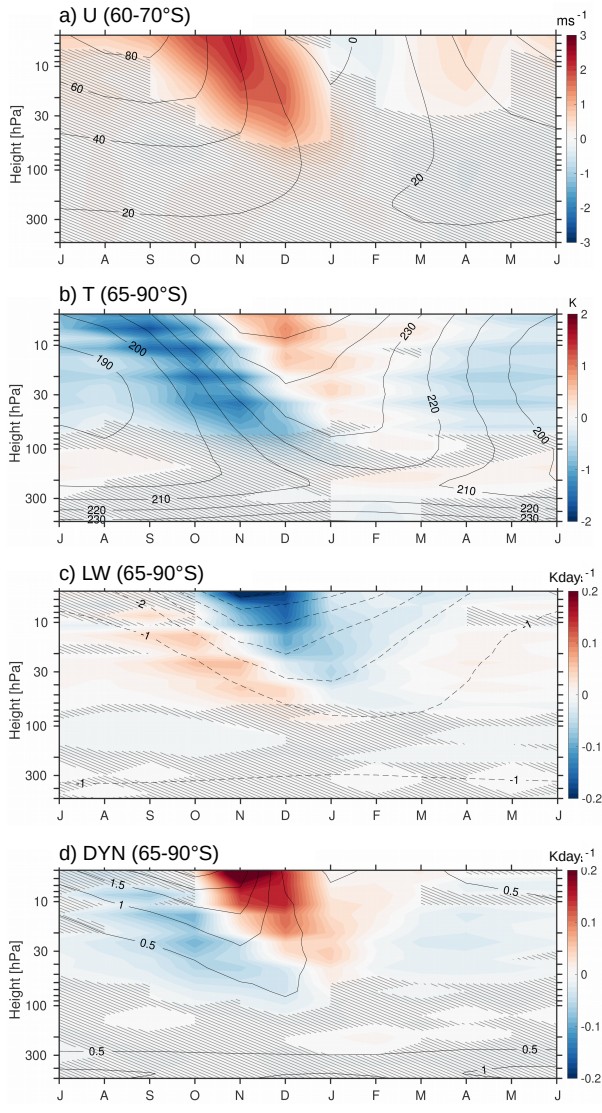

**Figure 2.** SH ensemble mean differences between Chem ON and Chem OFF for the climatological zonal mean zonal wind (U) in ms$^{-1}$ (a), polar cap temperature (T) in K (b), long–wave heating rate (LW) in K day$^{-1}$ (c), and dynamical heating rate (DYN) K day$^{-1}$ (d) as a function of height (shading). Contours represent the climatology of Chem ON. The contour intervals are 20 ms$^{-1}$ (a), 10 K (b), 1 K day$^{-1}$ (c), and 0.5 K day$^{-1}$ (d). Statistically insignificant regions are hatched at the 5% level based on a two-sample t-test.

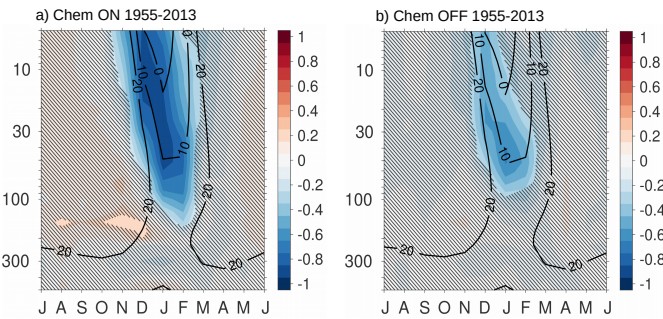

**Figure 3.** Correlations between ozone at 50 hPa and dynamical heating rates for (a) Chem ON and (b) Chem OFF as a function of height for 1955 to 2013 (shading). The particular climatological zonal mean zonal wind is represented for values $\leq 20$ ms$^{-1}$ for the same period with an interval of $10$ ms$^{-1}$. In the non-hatched area more than half of the ensemble members (at least 5 out of 9 members) show significant correlation coefficients with p-values $\leq 0.05$.

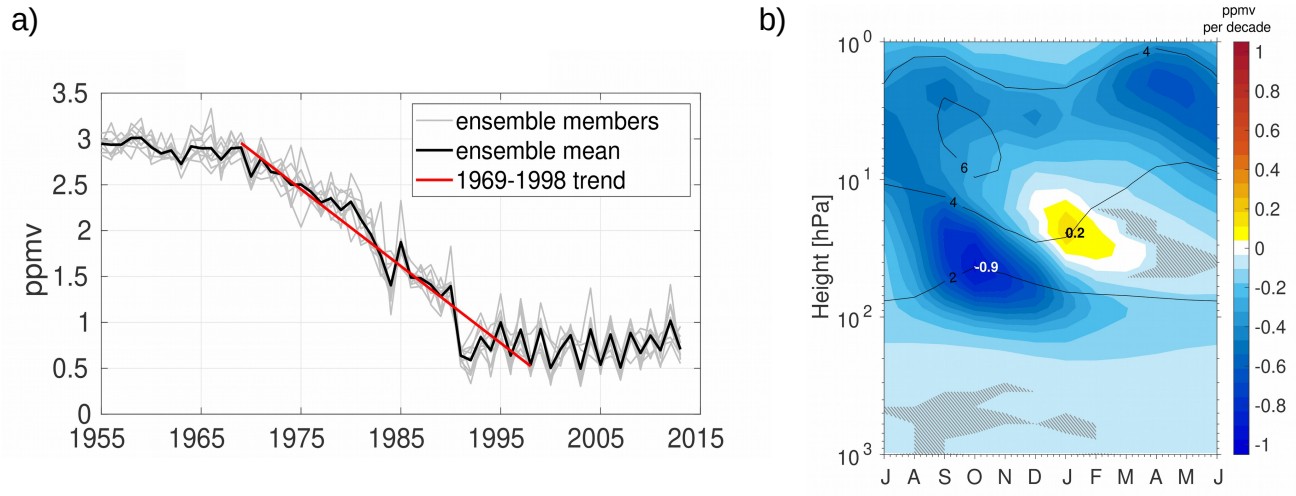

**Figure 4.** (a) Polar cap (65°to 90°S) ozone time series for October at 50 hPa for the single ensemble members (grey) and the ensemble mean of Chem ON (black). The red line depicts the linear trend in ozone from 1969 to 1998 in Chem ON. Please note that the ozone time series shown here for Chem ON is the same for Chem OFF and for Chem OFF 3D. (b) Polar cap (65°-90°S) linear ozone trend in $\mathrm{ppmv\,decade^{-1}}$ as function of height for the ensemble mean of Chem ON for the time period 1969-1998 (shading). The climatology in ozone (contours) is represented for the same period with an interval of 1 ppmv. Statistically insignificant trends are hatched at the 5% level based on a Mann-Kendall test.

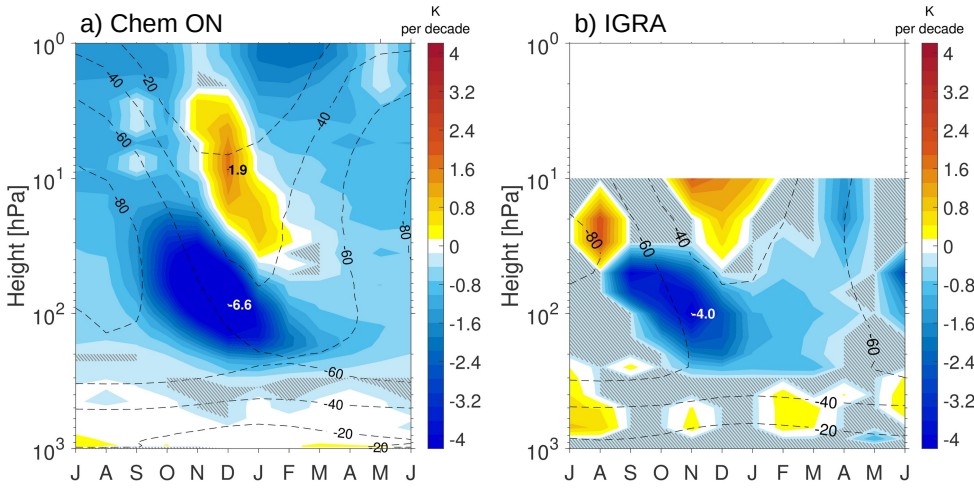

**Figure 5.** Polar cap (65°-90°S) linear temperature trend in $K \, \text{decade}^{-1}$ as function of height for the ensemble mean of (a) Chem ON and (b) IGRA for the time period 1969-1998 (shading). The particular climatologies (contours) are represented for the same period with an interval of 20 K. Statistically insignificant regions are hatched at the 5% level based on a Mann-Kendall test.

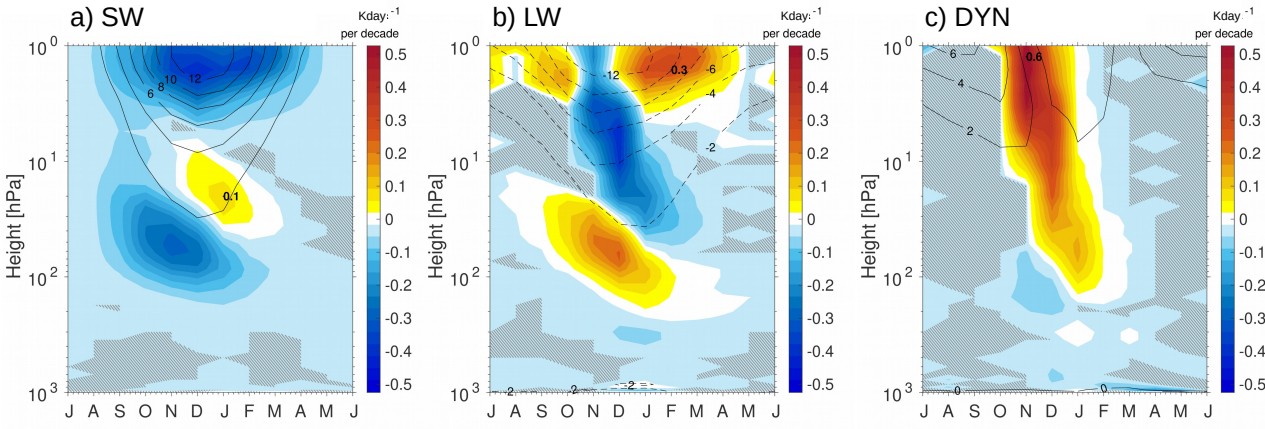

**Figure 6.** Polar cap (65°-90°S) linear (a) SW, (b) LW and (c) DYN heating rate trends in $\mathrm{K\,day^{-1}\,decade^{-1}}$ as function of height for the ensemble mean of Chem ON for the time period 1969-1998 (shading). The particular climatologies (contours) are represented for the same period with an interval of $2\,\mathrm{K\,day^{-1}}$. Statistically insignificant regions are hatched at the 5% level based on a Mann-Kendall test.

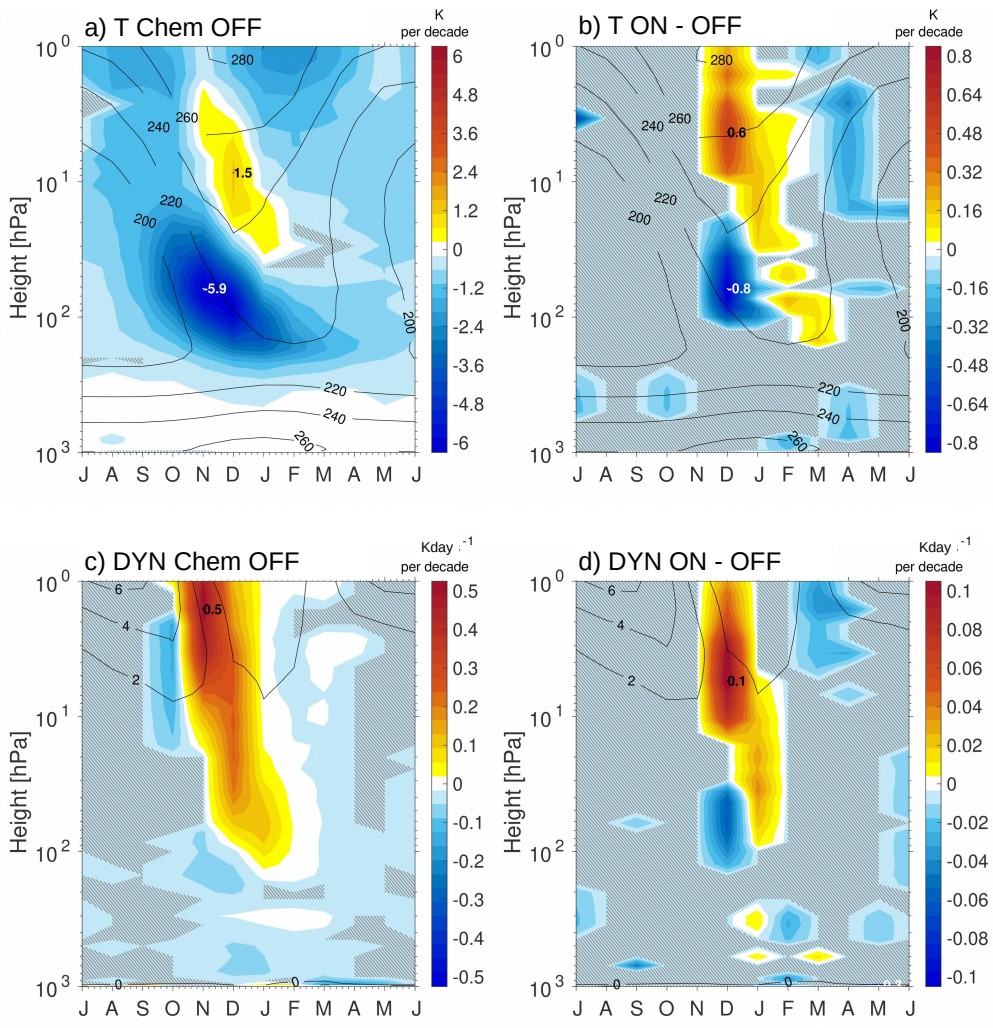

**Figure 7.** Polar cap (65°-90°S) linear (a) temperature trend (T) in $\mathrm{K\,decade^{-1}}$ and (c) dynamical heating rate trend (DYN) in $\mathrm{K\,day^{-1}\,decade^{-1}}$ as function of height for the ensemble mean of Chem OFF for the time period 1969-1998 and the difference to Chem ON (b, d). The particular climatologies (contours) are represented for the same period with an interval of 20 K (a, b) and $2\,\mathrm{K\,day^{-1}}$ (c, d). Statistically insignificant regions are hatched at the 5% level based on a Mann-Kendall test.

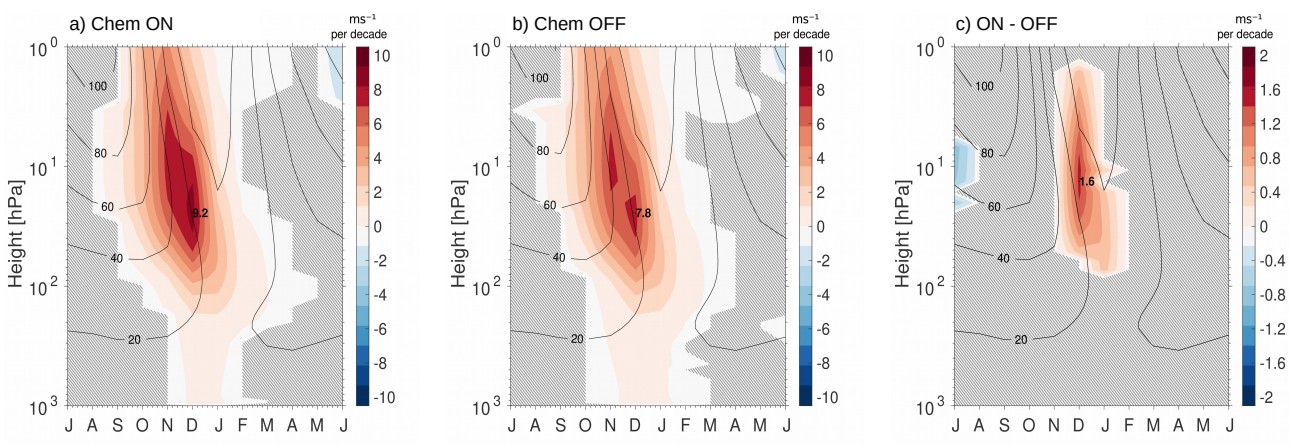

**Figure 8.** Zonal mean zonal wind trend (60°-70°S) in $\mathrm{ms^{-1}\,decade^{-1}}$ as function of height for the time period 1969-1998 (shading) for the ensemble mean of (a) Chem ON and (b) Chem OFF, as well as for (c) the differences (shading) between the simulations. The particular climatologies (contours) are represented for the same period with an interval of $20\,\mathrm{ms^{-1}}$. Statistically insignificant regions are hatched at the 5% level based on a Mann-Kendall test.

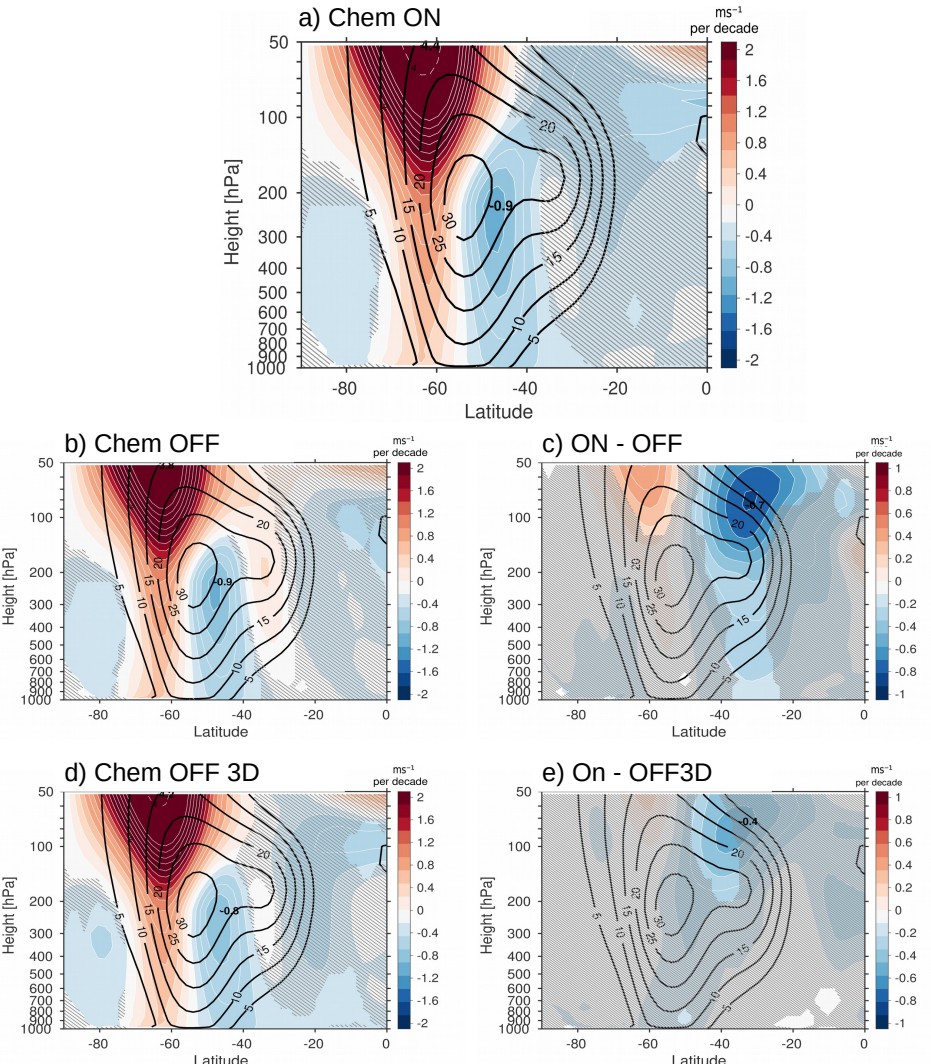

**Figure 9.** Zonal mean zonal wind trend in $\mathrm{ms^{-1}\,decade^{-1}}$ in the troposphere and lower stratosphere for 1969-1998 DJF (shading) for (a) Chem ON, (b) Chem OFF, and (d) Chem OFF 3D (d). The white contours represent values $\geq 3\ \mathrm{ms^{-1}\,decade^{-1}}$ with an interval of 0.2 $\mathrm{ms^{-1}\,decade^{-1}}$. The differences (shading) between the simulations are presented for Chem ON - OFF (c) and Chem ON - OFF 3D (e). The particular climatologies (contours) are represented for the same period with an interval of 5 $\mathrm{ms^{-1}}$. Statistically insignificant regions are hatched at the 5% level based on a Mann-Kendall test.

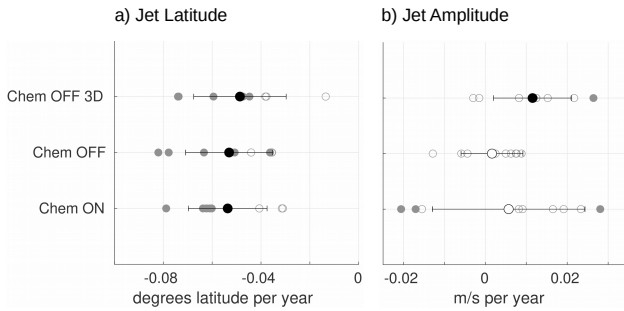

**Figure 10.** 1969-1998 DJF Trend for the 850 hPa a) jet latitude (in degrees latitude per year) and b) jet amplitude (in $\mathrm{m\,s^{-1}}$ per year) in the different model experiments. Single ensemble members are shown in gray; the ensemble mean is shown in black including an error bar for one standard deviation. Filled circles show a statistically significant trend at the 95% level based on a Mann-Kendall test.

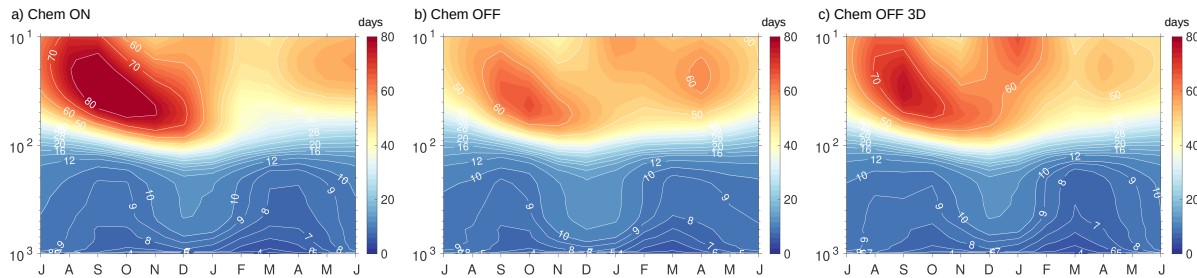

**Figure 11.** SAM timescale in days for the time period 1955-2013 for the (a) Chem ON, (b) Chem OFF, and (c) Chem OFF3D ensembles.

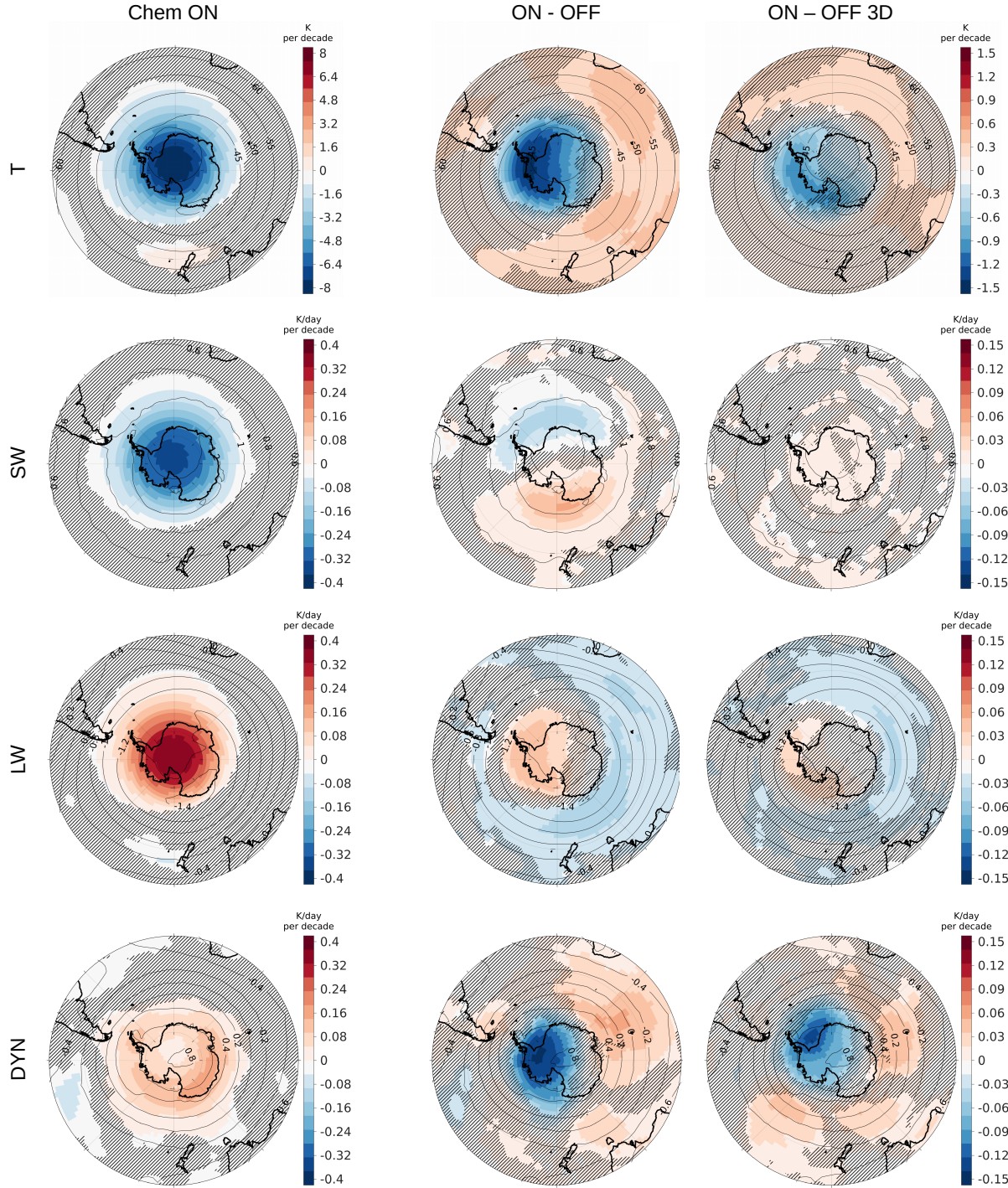

**Figure 12.** Temperature (T), short–wave heating rate (SW) , long–wave heating rate (LW), and dynamical heating rate (DYN) trends in $\mathrm{K\,decade^{-1}}$ and $\mathrm{K\,day^{-1}\,decade^{-1}}$ at 50 hPa for 1969-1998 in December (shading) for (a) Chem ON, (b) Chem ON minus Chem OFF, and (c) Chem ON minus Chem OFF 3D. The particular climatologies (contours) are represented for Chem ON during the same period with an interval of 5 K for T and $0.2\,\mathrm{K\,day^{-1}}$ for SW and DYN. Statistically insignificant regions are hatched at the 5% level based on a Mann-Kendall test.

**Table 1.** Different model settings of CESM1 used in this study and their respective abbreviations.

| Model Version | Ensemble Members | Years | Ozone Setting | Abbreviation |
|---|---|---|---|---|
| CESM1(WACCM) | 9 | 1955-2013 | Interactive | Chem ON |
| CESM1(SC-WACCM) | 9 | 1955-2013 | Daily zonally symmetric* | Chem OFF |
| CESM1(SC-WACCM) | 9 | 1955-2013 | Daily asymmetric* | Chem OFF 3D |

* The ozone data used for prescription originates from the Chem ON run.

**Table 2.** SH polar cap (65-90°S) magnitude and month of the maximum negative temperature trends for the time period (1969-1998) from different studies based on model and observational radiosonde data. The number of ensemble members is indicated in brackets for model simulations. The $2\sigma$ errors are also shown where available.

| Data | Trend in K decade$^{-1}$ | Month | Source |
|---|---|---|---|
| **Observed Trends** | | | |
| Radiosonde data [1] | -2.2 | Nov | Thompson and Solomon (2002) |
| IUK [2] | -4.7 ± 2.8 | Nov | Young et al. (2013) |
| RICH-obs [3] | -4.1 ± 2.4 | Nov | Young et al. (2013) |
| HadAT2 [4] | -3.8 ± 2.4 | Nov | Young et al. (2013) |
| IGRA | -4.0 | Nov | This work |
| **Modeled Trends** | | | |
| WACCM4 (3) [5] | -6.7 ± 3.0 | Dec | Calvo et al. (2017) |
| WACCM-CCMI (3) [6] | -4.4 ± 2.8 | Nov | Calvo et al. (2017) |
| Chem ON (9) | -6.6 ± 1.4 | Dec | This work |
| Chem OFF (9) | -5.9 ± 2.0 | Nov | This work |
| Chem OFF 3D (9) | -6.3 ± 0.6 | Dec | This work |

[1] Stations: SANAE, Halley, Syowa, Molodeznaja, Davis, Mirnyj, Casey

[2] Iterative Universal Krigin

[3] Radiosonde Innovation Composite Homogenization, version 1.5

[4] Hadley Centre Atmospheric Temperatures, version 2

[5] WACCM, version 4

[6] WACCM-Chemistry-Climate Model Initiative