# Peer review of "Sensitivity of the southern hemisphere circumpolar jet response to Antarctic ozone depletion: prescribed versus interactive chemistry"

_Atmospheric Chemistry and Physics, 2020_

## Referee Comment (RC1) · Anonymous Referee #1 · 20 Jun 2020

This manuscript examines the impact of the representation of stratospheric ozone on climate model simulations of tropospheric jet trends, by comparing ensembles of simulation with (i) interactive chemistry, (ii) prescribed zonal-mean ozone, and (iii) prescribed 3D ozone. This is an important topic that is relevant for ACP, and the manuscript is generally well written and presents some new results. However, before it can be published there needs to be more, quantitative analysis of the differences in jet trends among simulations, as well as discussion of some relevant previous studies that have not been cited.

MAJOR COMMENTS

1. The title indicated that the tropospheric jet is the focus of this study, but most of the focus is on the stratosphere and not the troposphere. Only one subsection of results is on tropospheric jet, only 2 out 9 figures show the tropospheric jet, and the first 1.5 pages of Introduction are on stratosphere and only at line 85 is surface/tropospheric features discussed. I think there should be more discussion and analysis of the tropospheric jet, and less material on stratospheric changes.

2. Regarding additional analysis, there are statements on how the shift in the jet differs between the ON, OF and OFF 3D runs (lines 364-370 and 435) but this is not quantified. The near-surface differences shown in the fig 9c and e and small (and generally insignificant), and it is not clear from these plots how different the jet trends are. As the tropospheric jet response is the focus not the paper trends in the latitude and strength of the tropospheric jet (e.g. u at 850 hPa) need to be calculated, and compared between different model runs (as well as reanalyses). Do the trends differ, and how large is the difference compared to model-data differences? This is important given the comment on lines 3 and 22 in the abstract (see minor comment 1), and also Seviour et al. (2017) (Major Comment 3).

3. Missing references

Several key references are missing.

Waugh et al. 2009 was one of the first (if not the first) studies to look at impact of interactive versus specific ozone on SH trends, and should be included at least in discussion on pg 4)

Seviour et al. 2016 compared runs with specified daily and month ozone (see, in particular, section 3b) and should be referenced. See also minor comment 2.

Seviour et al. 2017 compared different simulations of the tropospheric response to ozone depletion (including the results from the 2016 paper). This paper showed that the statistical uncertainty in tropospheric jet changes was very large, and although

there were variations among simulations in the mean changes they all agreed within their uncertainties. Is this also the case for the 3 ensembles considered here?

MINOR COMMENTS

1. Line 3: "differ largely" and Line 22 "crucial for representing" both appear to be over-statements, both based on previous studies and this study. Yes there are differences depending on the ozone but I am not sure can be classed as large or crucial.

2. Line 10: "In contrast to earlier studies, we use daily-resolved ozone fields". This is not the first study to use daily-resolved ozone (e.g. Neely et al, Seviour et al. 2016).

3. Line 390, 445: Iyvanciu et al. (in prep). At the very least a paper needs to be submitted before it can be referenced.

4. Line 410: I don't understand what is meant by "The LW heating rate trend does not add more information".

REFERENCES

Waugh, D. W., L. Oman, P. A. Newman, R. S. Stolarski, S. Pawson, J. E. Nielsen, and J. Perlwitz (2009), Effect of zonal asymmetries in stratospheric ozone on simulated Southern Hemisphere climate trends , Geophys. Res. Lett., 36, L18701, doi:10.1029/2009GL040419.

Seviour, W.J.M, A. Gnanadesikan, D.W. Waugh, 2016: The Transient Response of the Southern Ocean to Stratospheric Ozone Depletion, J. Climate, 29, 7383-7396.

Seviour, W.J.M, D. W. Waugh, L. M. Polvani, G.J. P. Correa, C.I. Garfinkel, 2017: Robustness of the simulated tropospheric response to ozone depletion, J Climate, , 30, 2577-2585. doi.org/10.1175/JCLI-D-16-0817.1

---

## Referee Comment (RC2) · Susan Solomon (Referee) · 27 Jun 2020

This is an interesting and timely paper attacking an important problem. The findings are novel and certainly merit publication. I do have a number of questions and comments that I hope the authors find helpful in revising their paper. I don't think these suggestions are necessary since the paper is already quite good, but I do think they may make it clearer and stronger.

Substantive comments 1) The paper does a very good job on probing stratospheric change, but doesn't cover the tropospheric linkages as clearly. WACCM's Antarctic sea ice retreat has long been an issue in this (see Arblaster et al. recent paper) so it may

be necessary to say that it's not a good model to study the problem, but linkages still should show up better in the data presented for comparison, and I am puzzled by that. Much more could be done (for example, do you think sea ice and surface temperature trends should be further discussed?), but at least what is shown should be clear. I am surprised that in Figure 5, IGRA was chosen for the temperature comparison; there are now rather better databases out there including ERA5 and MERRA2. I am also very surprised that the temperature trend does not penetrate into the troposphere in January in Fig 5 and 7, can you please discuss/explain. Since there are linkages seen in zonal winds shown in Figures 8 and 9, I am very puzzled. Perhaps it's down to choice of latitude range? Not clear to me. Also, such linkages are often clearer when geopotential height is plotted rather than temperature, and that might be considered. I think the paper needs a clearer bottom line on whether feedbacks matter or do not matter for the tropospheric response, or whether this model's poor simulation of sea ice changes means that it's not suitable for such testing and that the troposphere is therefore not the focus here.

2) Total ozone is not a very sensitive diagnostic for evaluating a model's ability to simulate the coupling of interest here (Fig 4). Would it not be more useful to show some observations for comparison to Fig 4b instead. You could use the BDBP dataset or SWOOSH. It would be important to assess the vertical profile of the losses; much more so than the total column. Also, I would suggest that rather than showing the trends in ppmv per decade that you show the total trend over the period in percent. That is because what we really want to know is whether we see near complete depletion over the region from about 15-25 km at the peak, which gives us a good sense of the model's performance.

3) It is well known that there is a heating rate issue in the way that the SC version of WACCM handles mesospheric ozone; in particular, diurnal effects are not correctly accounted for and there are spurious rates of heating as soon as you get up above about 2 mbar, where atomic oxygen and ozone interchange from day into night and

drive a big diurnal variation that will give you an incorrect 24-hour average heating rate if you do not take account that SW heating is only present in daytime; this is all discussed in detail in Smith et al. Please comment on whether this may influence some of your results, particularly up near the 1 mbar level, and whether that has any potential to propagate downwards through a corresponding error in the residual circulation.

4) Lin et al. (J. Clim., 2009) showed evidence for a seasonal shift in the location of the polar vortex from July to November in the lower stratosphere. You have the perfect setup to test whether this occurs similarly irrespective of feedbacks and non-zonal forcing, and its relationship to BDC changes. It would be easy for you to reproduce their Figures 2-4, for your different cases; note the comparison to reanalysis shown in their Fig 8. It might help in understanding further what is going on in your Figure 11.

5) You emphasize the jet but it would be helpful to have more detail on the changes. It would be nice to make a simple scatter plot of the poleward shift of the jet (degrees) in the several simulations for various months, with the different simulations on the y axis and the observations on the x axis, or possibly using months on the x axis and plotting both data and models in the heart of the jet core on the y axis. We need to be able to see how many degrees the shift is in a simple way.

Minor comments

6) line 29 The paper's science is great but in several places the English is rather clumsy. Can this be checked over by a technical editor in one of the authors' institutions? Language like "The annually reoccurring depletion in polar stratospheric ozone was tremendous" distracts from an otherwise excellent paper. The annually reoccurring depletion in polar stratospheric ozone was striking or was of interest to scientists, the public, and policymakers alike, etc. would be better. 7) line 30 Political action was taken to ban the responsible substances (termed: ozone depleting substances, ODSs) under the Montreal Protocol in 1987 is incorrect. Political action was begun that ultimately led to a ban on the responsible substances (termed: ozone depleting substances, ODSs)
* * *
Interactive
comment

under the Montreal Protocol in 1987. The original Protocol in 1987 did not mandate a ban, only a freeze on emission at then-current rates. 8) line 37 Need a reference to the first paper showing this by Shine (GRL, 1986). 9) line 103 Please clarify what the shortcoming of Rae et al. is; this is not very clear here. How does it work and why doesn't it capture heterogeneous loss?

---

## Referee Comment (RC3) · Anonymous Referee #3 · 30 Jun 2020

This paper reported simulations of the climate responses to ozone depletion by WACCM, and compared the ones with interactive chemistry schemes with those pre-scribing zonal and 3d daily ozone. Each experiment consists of 9 ensembles with fully coupled ocean. It is found that interactive chemistry produces stronger stratospheric cooling, stronger strengthening of the polar night jet, and stronger poleward shift of the tropospheric jet, despite of the identical changes in ozone and shortwave heating rates. The authors attribute the difference to a chemistry-dynamics feedback that is absent when ozone is prescribed. This work highlights the importance to include the interactive chemistry into the climate simulations, and sheds light on understanding the complex coupling between the stratospheric ozone and the climate system. The paper

is logically organized and well written. I am not fully convinced by the mechanisms the authors provided to explain the "chemistry-dynamics feedback", but I do think the results deserve publication. Below is my detailed comments:

1. The authors suggested that there is both positive and negative feedbacks between ozone changes and dynamics, which occurs at different seasons and levels, which involves the background zonal wind condition and wave-mean flow interaction. However, it is not clear why interactive chemistry and specified chemistry would behave different based on this mechanism. Both of them have the same changes in ozone and SW heating rates, then the initial changes in the zonal winds should also be similar since that simply follows the thermal wind balance. Wave-mean flow interaction would also work in a similar fashion in the two experiments.

2. The mechanism for the "positive feedback" over the lower stratosphere in Nov/Dec is especially unclear, which is a key component to explain the stronger cooling seen in the interactive chemistry simulations. If the authors are referring to the positive correlations in Fig. 3a, these positive correlations are very weak and not statistically significant.

3. The climatology is calculated as the averaged over 1955-2013. But because of the strong trend related to ozone depletion, a large portion of the difference in the climatology may be a reflection of the difference in trends. It might be more meaningful to compare climatology over the pre-depletion era, such as 1955-1970.

4. Changes in the tropospheric jet. As shown in Fig. 9, the difference between Chem on and off is mainly between 20S-40S, which is more equator-ward than the equator flank of the tropospheric jet. I doubt the wind anomalies there can affect the location of the jet. Can the authors calculate the latitudes of the surface jets and verify if the latitude actually differ between the two? It is also not clear how the ozone anomalies in the polar regions affect the tropospheric jets in the subtropics.

4. Line 301-303: The first sentence here seems to suggest both the model used here and WACCM4 have stronger trends than WACCM-CCMI. But the second sentence

suggests the opposite.

---

## Referee Comment (RC4) · Anonymous Referee #2 · 1 Jul 2020

Summary: I quite liked reviewing this paper. The authors systematically compared two configurations of the same chemistry-climate model (CESM1-WACCM plus some modifications) that differ in that one configuration has fully interactive ozone chemistry, whereas the other uses prescribed ozone fields generated by the same model. The authors also test the sensitivity to prescribing zonally symmetric versus zonally resolved ozone fields. They find that the two model configurations produce qualitatively similar results but with important quantitative differences, e.g. regarding the coupling of the polar vortex strength with ozone depletion, timescales of variability of the Southern Annular Mode, an acceleration of the westerlies in the Southern Hemisphere, etc. They find that prescribing zonally resolved ozone produces results that are generally closer

to those produced using interactive ozone.

The results are of interest to climate modellers weighing up whether to include interactive ozone in climate projection simulations. Often the additional computational cost and scientific effort needed to sustain this functionality are considered prohibitive. There is a small number of other papers that characterize the advantages of interactively simulating ozone, and often these papers involve comparisons that are not entirely balanced, e.g. by comparing groups of different models. There is thus clearly a niche for this paper that aims to quantify the differences between the two approaches with minimal interference from other model differences (that are not ozone). (The authors do not divulge any details about the computational costs of their three ensembles; maybe this small detail can be added.)

In a few places error bounds should be stated before an explanation is given for why two quantities are different. Otherwise we cannot be sure that such differences are not coincidental in nature. Haase and Matthes (2019) are cited extensively. Perhaps the authors could elaborate a bit more how the results produced here compare to the results shown in that reference. Do the differences come down to the same mechanism in both hemispheres?

I could not discern whether the model is in a coupled atmosphere-ocean or an atmosphere-only configuration. If it is coupled, do perhaps slow modes of oceanic variability influence the results (that perhaps evolve differently in the different ensembles)?

I also agree with another reviewer that the title should be revised given the balance of evidence presented in this paper. The SH tropospheric jet seems to be a relatively minor topic here. While the authors state that HM19 is about NH results, they are cited in reference to SH features too, so a bit more discussion about the key differences between the two papers would be good to have.

The language is mostly fine (some minor style issues are listed below), the figures are

informative and about right in number, the conclusions are balanced. I thus recommend publication in ACP once my comments are addressed.

Minor comments:

L5 and line 63: I suggest to replace "accurate" with "appropriate", "self-consistent" or similar. Eyring et al. (2013) showed for CMIP5 models that interactive ozone models can fail to produce "accurate" fields... (https://doi.org/10.1002/jgrd.50316)

L52: Whether ozone recovery will ever be stronger than GHG influences may also be a function of the assumed GHG scenario.

L62-72: A third, fairly widespread method is to use an online parameterization for ozone (i.e. make ozone interactive but not use comprehensive chemistry). This route is followed in several CMIP6 models, sometimes to the point that models with and without comprehensive-chemistry ozone schemes are almost indistinguishable in their performances (e.g. CNRM models). Given the results of this paper (showing that prescribing 3D ozone fields already constitutes progress) I'd say that for some groups this might be the way forward.

L89: Cut out "along".

L153: Replace "as well as" with "like".

L160: Replace "from the land fraction factor" with "on land fraction".

L162-163: How do you know the cold-pole bias has been reduced in pre-industrial simulations? Almost nothing is known about the pre-industrial stratosphere...

L236: Replace "It was shown in Haase and Matthes (2019)" with "HM19 showed".

L245: I don't understand why the amplitude of the response should be smaller with a larger ensemble. The mean response should be better defined as the ensemble size decreases. Please expand / explain.

L330: Whether these numbers are indeed different requires some kind of analysis of the statistical uncertainties. If they are not different within their uncertainty bounds, the argument would fall apart.

L373ff: Note also https://doi.org/10.1002/2014JD023009 who also studied stratospheric SAM variability and found an increase in variability under ozone depletion.

L390ff: I don't think it's appropriate here to discuss an unpublished paper. If it has meanwhile been published (but perhaps not fully peer-reviewed) that would be OK. If not, I suggest to remove this section.

L399: I'm sure the "polar stereographic" map projections are not relevant here, but rather the physical quantities / questions that you want to address. Which fields are you assessing here?

L447: Please spell out code availability, a requirement for publication in ACP.

---

## Referee Comment (RC5) · Anonymous Referee #4 · 29 Jul 2020

Review of "Sensitivity of the southern hemisphere tropospheric jet response to Antarctic ozone depletion: prescribed versus interactive chemistry" by Haase et al.

This paper investigates how Antarctic stratosphere and troposphere mean climate and climate change is affected by the model representation of Antarctic ozone. Three ensembles were performed for the 1955-2013 period using CESM with different ozone approach: interactive ozone, prescribed zonal-mean daily ozone, and prescribed 3D daily ozone. The results are consistent with previous studies that interactive ozone causes stronger Antarctic lower stratospheric cooling and stronger stratospheric jet response in austral summer.

This paper advances the understanding of the effects of interactive ozone on Antarctic climate change. Specifically, it emphasizes the role of reduced dynamical heating in causing Antarctic lower stratospheric cooling with interactive ozone. It also quantifies the impact of ozone-dynamics feedbacks and ozone zonal asymmetry on simulated temperature and jet trends. However, I think the authors' interpretation of how interactive ozone affects tropospheric jet trends needs to be clarified.

Major Comments:

I don't think the results presented in Figure 9 support the conclusion that interactive ozone leads to stronger poleward shift of the tropospheric jet. Figure 9 shows that interactive ozone does not significantly influence the tropospheric jet trends poleward of 40S. Significantly different tropospheric jet trends are only found between 20S and 40S (Fig. 9c), where the westerly trends are weaker in the interactive ozone simulations than in the prescribed ozone simulations. It would be useful to find out why interactive and prescribed ozone simulate different tropospheric subtropical jet trends. However, the major point is that the large stratospheric circumpolar jet differences between interactive and prescribed ozone simulations do not propagate downward into the troposphere.

Minor Comments:

The title should be changed because most of the paper is about the impact of interactive ozone on the SH stratosphere.

Lines 251-252: Why interactive ozone causes a weaker shallow BDC branch?

In some occasions, PNJ should be replaced with circumpolar jet.

---

## Author Comment (AC2) · 10 Sep 2020

Dear Susan,

thanks for your comments and suggestions on our paper. The detailed answer to your comment is included in our comment to the editor and all referees. Here, we only wanted to upload an additional supplement for you regarding the analysis of Lin et al. 2009. The PDF includes the analysis that Pu Lin did in her 2009 paper for figures 1 to 4 using our experiments.

Best Regards, Sabine

Please also note the supplement to this comment:
https://acp.copernicus.org/preprints/acp-2020-441/acp-2020-441-AC2-supplement.pdf

**Supplement:**

**SH lower stratospheric temperature and ozone climatologies and trends**

September 10, 2020

**Contents**

This PDF includes figures of temperature and ozone trends at 50 hPa in CESM1-WACCM with interactive chemistry (Chem ON), and specified chemistry using SC-WACCM as the atmosphere component with zonal mean forcing (Chem OFF) and zonally asymmetric forcing for the ozone concentrations (Chem OFF 3D), respecitively.

The analysis follows the work by Lin et al. 2009 (their Figures 1 - 4), but uses the 50 hPa level instead of the T4 weighted averages. Contours are included in all figures. Values are given in ppm for ozone and Kelvin for temperature. For areas with missing color, the values were out of range of the colorbar. For the trend plots a signifiance test was carried out following the Mann-Kandell test procedure for 95% significance. Insignificant areas are hatched. In case there is no hatching at all - the whole trend signal is insignificant.

**1 Climatology 1979-2007**

**1.1 Ozone at 50 hPa - total field**

[Figure]

**1.2   Ozone at 50 hPa - eddy field**

[Figure]

**1.3   Temperature at 50 hPa - total field**

[Figure]

**1.4 Temperature at 50 hPa - eddy field**

[Figure]

**2  Trend 1979-2007**

**2.1  Ozone at 50 hPa - total field**

[Figure]

**2.2  Ozone at 50 hPa - eddy field**

[Figure]

**2.3  Temperature at 50 hPa - total field**

[Figure]

**2.4  Temperature at 50 hPa - eddy field**

---

## Author Response (AR1)

Dear Martin Dameris, dear referees,

We would like to thank you for your very valuable suggestions and comments. All reviewers commented on the fact that we do not cover the tropospheric jet response sufficiently according to the title of the paper. We give a general answer to the editor and all reviewers about this issue in the beginning and give detailed answers to the specific points of the different reviewers highlighted in blue below.
At the end of this document, a revised version of the manuscript is attached. All changes made in this manuscript are highlighted in yellow.

Best regards,
Sabine Haase on behalf of all authors

**General answer to the editor and all referees**

Most referees are concerned that the tropospheric jet response to ozone depletion under the different chemistry settings is not addressed sufficiently with respect to the title. Furthermore, the statistical significance of the chemistry impact on the tropospheric jet was questioned.

We agree with the reviewers and to resolve these issues we changed the title as recommended by some of the referees, included an additional figure that evaluates the difference in the trend of the tropospheric jet strength and latitude (new Fig. 10), and adapted the manuscript text accordingly.

The title does not refer to the tropospheric jet in particular anymore and the new analysis shows that the difference in the shift of the tropospheric jet between interactive and specified chemistry settings is not statistically significant. We adapted the conclusion and discussion of our results accordingly.

Abstract:
- Line 14: "This difference between interactive and specified chemistry in the stratospheric response to ozone depletion also affects the tropospheric response. However, an impact on the poleward shift of the tropospheric jet stream is not detected."

Results:
- Line 395: "Figure 10 shows the trend for the tropospheric jet latitude and strength at 850 hPa. There is no statistically significant difference between the chemistry settings in the trend of the tropospheric jet position and strength. All experiments have a similar mean jet latitude trend and there is a large spread among ensemble members in the trend of the jet strength, which leads to hardly significant trends in the ensemble means. Therefore, the impact of interactive chemistry that is significant in the stratosphere does not seem to show the same significance in the troposphere."
- Line 418: "[…] the too short tropospheric SAM timescale in WACCM, which is found in all our experiments independent of the chemistry setting (Fig. 11), indicates that the coupling between the stratosphere and troposphere is very likely too weak. This could explain why we do not find significant differences in the tropospheric jet trends between our experiments (Fig. 10) […]."

Conclusions:
- Line 493: "However, the impact of interactive chemistry on the tropospheric jet could not be validated by our study. […] Although not directly affecting the position of the tropospheric jet, the differences we find between the chemistry settings (Fig. 9), show a stronger tropospheric response to ozone depletion when interactive chemistry is included. An updated model version of WACCM, based on the CAM5 physics, might improve our understanding of the stratospheric impact onto the troposphere under different chemistry settings."

**Anonymous Referee #1**

This manuscript examines the impact of the representation of stratospheric ozone on climate model simulations of tropospheric jet trends, by comparing ensembles of simulation with (i) interactive chemistry, (ii) prescribed zonal-mean ozone, and (iii) prescribed 3D ozone. This is an important topic that is relevant for ACP, and the manuscript is generally well written and presents some new results. However, before it can be published there needs to be more, quantitative analysis of the differences in jet trends among simulations, as well as discussion of some relevant previous studies that have not been cited.

**MAJOR COMMENTS**

1. The title indicated that the tropospheric jet is the focus of this study, but most of the focus is on the stratosphere and not the troposphere. Only one subsection of results is on tropospheric jet, only 2 out 9 figures show the tropospheric jet, and the first 1.5 pages of Introduction are on stratosphere and only at line 85 is surface/tropospheric features discussed. I think there should be more discussion and analysis of the tropospheric jet, and less material on stratospheric changes.

We agree that the title was not chosen wisely since a large part of the paper is focusing on the stratosphere. We also agree that we valued the impact of interactive chemistry onto the tropospheric jet too much. We adapted these sections in the paper (see also your minor comments) and included an additional Figure on the 850 hPa trends of the jet stream in the revised manuscript (new Fig. 10). This additional analysis to better describe the impact of interactive chemistry on the jet stream in the troposphere is presented below (answer to major comment 2).
However, we do not want to cut on the stratospheric part of the paper since this is the region in which the strongest differences occur and which is needed to understand the mechanism.

2. Regarding additional analysis, there are statements on how the shift in the jet differs between the ON, OF and OFF 3D runs (lines 364-370 and 435) but this is not quantified. The near-surface differences shown in the fig 9c and e and small (and generally insignificant), and it is not clear from these plots how different the jet trends are. As the tropospheric jet response is the focus not the paper trends in the latitude and strength of the tropospheric jet (e.g. u at 850 hPa) need to be calculated, and compared between different model runs (as well as reanalyses). Do the trends differ, and how large is the difference compared to model-data differences? This is important given the comment on lines 3 and 22 in the abstract (see minor comment 1), and also Seviour et al. (2017) (Major Comment 3).

As proposed we calculated the jet latitude and strength at 850 hPa. We followed the procedure as described in Seviour et al. (2017), defining the jet latitude and strength as the location of the maximum of a quadratic fitted to the 850 hPa zonal mean zonal wind at its maximum grid point and the two points either side.

[Figure]

*Figure A 1: 850hPa jet latitude and amplitude difference between 1958-1968 and 1995-2005 for the different chemistry settings in CESM1-WACCM. The dashed line indicates the result for ERA data (combination of ERA-40 and ERA-Interim). The ensemble members are shown in gray while the ensemble mean is shown in black. Solid circles indicate significance at the 95% level using a t-test.*

Figure A 1 shows the tropospheric jet latitude and trend as the difference between the averages over the periods 1958-1968 and 1995 to 2005 for a better comparison to Seviour et al. 2017. The three different WACCM experiments basically agree on the mean trend in jet latitude and strength. The differences are statistically not significant (not shown). Compared to the multi-model mean presented in Seviour et al. 2017, the trends for our WACCM simulations are stronger for the jet latitude and weaker for the jet amplitude but well in between the spread among the different models used in Seviour et al. 2017. The comparison to a combined data set using ERA-40 and ERA-Interim data shows that WACCM rather underestimates the trends in the strength of the tropospheric jet as well as in the jet latitude although the trend in jet latitude is better captured than the trend in the strength of the tropospheric jet.

For the paper, we decided to use the linear regression method (Fig. A 2) as we did for the other figures. The main result is the same as in Figure 1 A. There is no significant difference between the chemistry settings considering the trend of the jet position or strength. Using a linear regression, though, reduces the significance of the ensemble mean trend for the strength of the tropospheric jet (open circles in Fig. A 2 as compared to filled ones in Fig. A 1), due to the large spread among ensemble members.

[Figure]

*Figure A 2: Same as Figure A 1, but using a linear regression to determine the trend for the period 1969 to 1998 as done in the paper. Significance is detected using a Mann-Kandell Test.*

3. Missing references. Several key references are missing. Waugh et al. 2009 was one of the first (if not the first) studies to look at impact of interactive versus specific ozone on SH trends, and should be included at least in discussion on pg 4) Seviour et al. 2016 compared runs with specified daily and month ozone (see, in particular, section 3b) and should be referenced. See also minor comment 2. Seviour et al. 2017 compared different simulations of the tropospheric response to ozone depletion (including the results from the 2016 paper). This paper showed that the statistical uncertainty in tropospheric jet changes was very large, and although there were variations among simulations in the mean changes they all agreed within their uncertainties. Is this also the case for the 3 ensembles considered here?

We apologize for missing these references in the first version of the manuscript and include them now in the introduction of the revised manuscript.

Seviour et al. 2016:
Line 98: "Seviour et al. (2016) showed that using a daily ozone forcing does not only increase the effect of ozone depletion on the atmospheric response but that an impact is also found in the interior of the ocean."

Seviour et al. 2017:
Line 123: "In another multi--model study, Seviour et al. (2017) argue that interannual variability is very strong and large ensembles or long time slice simulations are required to detect robust differences among models regarding the signal in the troposphere from stratospheric ozone depletion."

Waugh et al. 2009:
Line 135: "One of the first studies addressing this issue was carried out by Waugh et al. (2009). Using NASA's Goddard Earth Observing System Chemistry-Climate Model (GEOS CCM) to investigate the effect of SH ozone trends on the atmospheric circulation, they found a stronger

cooling (warming) trend in the stratosphere for ozone depletion (recovery) with interactive chemistry and an underestimation of Antarctic temperature trends and trends in the SAM when ozone was prescribed as a monthly mean in the CCM. Li et al. (2016) confirmed the results from Waugh et al. (2009) coupling version 5 of the same CCM (GEOS-5) to an interactive ocean. […]"

MINOR COMMENTS

- Line 3: "differ largely" and Line 22 "crucial for representing" both appear to be overstatements, both based on previous studies and this study. Yes there are differences depending on the ozone but I am not sure can be classed as large or crucial.

  We rephrased these sentences a follows:
  - […] differ among climate models […]
  - […] could also have an influence on the representation of […]

- Line 10: "In contrast to earlier studies, we use daily-resolved ozone fields". This is not the first study to use daily-resolved ozone (e.g. Neely et al, Seviour et al. 2016).

  We deleted "In contrast to earlier studies" here and included the Seviour study in the introduction of the revised manuscript.

- Line 390, 445: Iyvanciu et al. (in prep). At the very least a paper needs to be submitted before it can be referenced.

  The paper is now submitted and will shortly be a discussion paper in ACP as well. We adapted the reference accordingly.

- Line 410: I don't understand what is meant by "The LW heating rate trend does not add more information".

  We meant that the LW heating rate is only a response to the dynamical heating and the SW heating and that it does not add more knowledge to why the patterns differ between Chem ON and Chem OFF. To avoid confusion here, we rephrased the sentence by deleting "does not add more infromation".

  "The LW heating rate trend (Fig. 12) dampens the signal from the dynamical heating rate trend."

Anonymous Referee #2

Summary: I quite liked reviewing this paper. The authors systematically compared two configurations of the same chemistry-climate model (CESM1-WACCM plus some modifications) that differ in that one configuration has fully interactive ozone chemistry, whereas the other uses prescribed ozone fields generated by the same model. The authors also test the sensitivity to prescribing zonally symmetric versus zonally resolved ozone fields. They find that the two model configurations produce qualitatively similar results but with important quantitative differences, e.g. regarding the coupling of the polar vortex strength with ozone depletion, timescales of variability of the Southern Annular Mode, an acceleration of the westerlies in the Southern Hemisphere, etc. They find that prescribing zonally resolved ozone produces results that are generally closer to those produced using interactive ozone. The results are of interest to climate modellers weighing up whether to include interactive ozone in climate projection simulations. Often the additional computational cost and scientific effort needed to sustain this functionality are considered prohibitive.

There is a small number of other papers that characterize the advantages of interactively simulating ozone, and often these papers involve comparisons that are not entirely balanced, e.g. by comparing groups of different models. There is thus clearly a niche for this paper that aims to quantify the differences between the two approaches with minimal interference from other model differences (that are not ozone). (The authors do not divulge any details about the computational costs of their three ensembles; maybe this small detail can be added.)

In a few places error bounds should be stated before an explanation is given for why two quantities are different. Otherwise we cannot be sure that such differences are not coincidental in nature.

Haase and Matthes (2019) are cited extensively. Perhaps the authors could elaborate a bit more how the results produced here compare to the results shown in that reference. Do the differences come down to the same mechanism in both hemispheres?

I could not discern whether the model is in a coupled atmosphere-ocean or an atmosphere-only configuration. If it is coupled, do perhaps slow modes of oceanic variability influence the results (that perhaps evolve differently in the different ensembles)?

I also agree with another reviewer that the title should be revised given the balance of evidence presented in this paper. The SH tropospheric jet seems to be a relatively minor topic here. While the authors state that HM19 is about NH results, they are cited in reference to SH features too, so a bit more discussion about the key differences between the two papers would be good to have.

The language is mostly fine (some minor style issues are listed below), the figures are informative and about right in number, the conclusions are balanced. I thus recommend publication in ACP once my comments are addressed.

Thank you for your comments and suggestions. As described above, we adapted the title and included additional analysis on the tropospheric jet trend, specifically the trends in 850 hPa jet latitude and strength.

It is stated in the manuscript that we use the fully-coupled model version (e.g., lines 8 and 182 in the first version of the manuscript). To make this clearer, we now explicitly state that an interactive ocean is included.

- Line 164: "CESM1(WACCM) is a fully coupled climate model with interactive ocean, land and sea ice components."
- Line 196: "All simulations were performed in a fully–coupled setup with the same interactive ocean, land and sea ice components."

Possible impacts of the ocean on our results are now mentioned in the discussion part of the paper. We suggest that these impacts are low following the findings of Gillett et al. 2019, but cannot exclude that SST and sea ice biases might feed back onto the position of the tropospheric jet stream.

- Line 493: "However, the impact of interactive chemistry on the tropospheric jet could not be validated by our study. This might be due to the weak stratosphere-troposphere-coupling in the model that is indicated by the low tropospheric time scale of the SAM. This feature might be connected to the interactive ocean, which shows large biases in sea ice retreat in the seasonal cycle (Landrum et al. 2012; Marsh et al. 2013). However, a recent study by Gillett et al. (2019) showed that the response between ozone depletion and the SAM was independent from coupling an interactive ocean to WACCM or running it with observed SSTs."

We included an additional sentence on the NH paper (Haase and Matthes 2019), but otherwise think it is sufficiently covered already.

- Line 89: "They found especially the negative feedback at the end of the winter season to be important for the difference between specified and interactive chemistry simulations, which led to a more rapid and earlier stratospheric vortex break-down in the interactive chemistry simulations."

We now include a remark on the computational costs of the different WACCM settings. Thank you for this suggestion!

- Line 209: "The specified chemistry setup runs about 4 times faster than the full chemistry setup and is therefore computationally much cheaper."

Minor comments:

L5 and line 63: I suggest to replace "accurate" with "appropriate", "self-consistent" or similar. Eyring et al. (2013) showed for CMIP5 models that interactive ozone models can fail to produce "accurate" fields: (https://doi.org/10.1002/jgrd.50316)

We now use "appropriate".

L52: Whether ozone recovery will ever be stronger than GHG influences may also be a function of the assumed GHG scenario.

Yes, indeed. We included this point in the sentence:
"However, when exactly ozone recovery is strong enough to compensate GHG cooling is an open question and also depends on future GHG levels."

L62-72: A third, fairly widespread method is to use an online parameterization for ozone (i.e. make ozone interactive but not use comprehensive chemistry). This route is followed in several CMIP6 models, sometimes to the point that models with and without comprehensive-chemistry ozone schemes are almost indistinguishable in their performances (e.g. CNRM models). Given the results of this paper (showing that prescribing 3D ozone fields already constitutes progress) I'd say that for some groups this might be the way forward.

Thank you for pointing this out. We now also mention this method in the introduction.
Line 106: "For example, an online parameterization or simplified online scheme for ozone can be applied. This is a step in between a fully-interactive and a specified chemistry setup and allows the ozone field to follow the dynamics to a certain degree, e.g., as in CNRM-CM6 (Voldoire et al. 2019) or E3SM-1-0 (Golaz et al. 2019)."

L89: Cut out "along".

Thanks for the suggestion. We changed the sentence accordingly.

L153: Replace "as well as" with "like".

Thanks for the suggestion. We changed the sentence accordingly.

L160: Replace "from the land fraction factor" with "on land fraction".

Thanks for the suggestion. We changed the sentence accordingly.

L162-163: How do you know the cold-pole bias has been reduced in pre-industrial simulations? Almost nothing is known about the pre-industrial stratosphere.

This sentence is referring to a model simulation only. We ran the two different WACCM configuration (with and without the changes applied to the model code as described in the paper) only under piControl conditions, unfortunately. We found that without the adaptations to the model code the stratospheric polar vortex is colder and stronger under this piControl conditions. We think it is appropriate to assume that under historical conditions a similar weakening of the polar vortex is achieved when applying the same modifications to the model code.

L236: Replace "It was shown in Haase and Matthes (2019)" with "HM19 showed".

We changed the structure of the sentence as suggested but did not use the abbreviation since it is introduced here.

L245: I don't understand why the amplitude of the response should be smaller with a larger ensemble. The mean response should be better defined as the ensemble size decreases. Please expand / explain.

Of course, it depends on which ensemble member you compare to the ensemble mean. Since the ensemble mean is an average, the individual members can show smaller or larger values at a specific position and time. On the NH, which is showing strong variability on the interannual scale individual members show strong differences, which differ only slightly in time or space. Averaging will result in an overall lower amplitude. Often features in the NH stratosphere are large in amplitude but small in significance since the impact of natural variability is dominating.
For the SH, this might not always be the case since the strong trend signal is very similar between the different individual members of an ensemble. For an example, we refer to the supplementary figure on the SAM timescale, which shows the individual ensemble members of the Chem ON simulation. Some of the individual model results show a much longer timescale than the ensemble mean shown in the main body of the paper (only one member shows a shorter time scale, three members agree in amplitude with the ensemble mean). There is also a disagreement in timing of the longest SAM timescale in the stratosphere between the ensemble members. Averaging these leads to a smaller amplitude but of course, to a better defined mean response. We slightly changed the wording in the manuscript saying that we do not find it unexpected to find a weaker response (line 266).

L330: Whether these numbers are indeed different requires some kind of analysis of the statistical uncertainties. If they are not different within their uncertainty bounds, the argument would fall apart.

Figure 7b shows that the trend difference between Chem ON and Chem OFF is significant starting in December. We think that is sufficient to prove our argumentation.

L373ff: Note also https://doi.org/10.1002/2014JD023009 (Dennison et al. 2015) who also studied stratospheric SAM variability and found an increase in variability under ozone depletion.

Thank you for pointing that out. We included this reference in the revised manuscript.
Line 403: "Dennison et al. (2015), for example, showed that under ozone depletion the SAM timescale is enhanced and stratosphere-troposphere-coupling is strengthened."

L390ff: I don't think it's appropriate here to discuss an unpublished paper. If it has meanwhile been published (but perhaps not fully peer-reviewed) that would be OK. If not, I suggest to remove this section.

The mentioned paper is submitted to ACPD now, too. We updated the reference.

L399: I'm sure the "polar stereographic" map projections are not relevant here, but rather the physical quantities / questions that you want to address. Which fields are you assessing here?

We revised this sentence as follows:
Line 433: "To better understand the improvement in the Chem OFF 3D ensemble over the Chem OFF ensemble we consider spatially asymmetric trends of temperature, SW, LW and dynamical heating rates in the following."

L447: Please spell out code availability, a requirement for publication in ACP.

We include this now in the dedicated section.

Anonymous Referee #3

This paper reported simulations of the climate responses to ozone depletion by WACCM, and compared the ones with interactive chemistry schemes with those prescribing zonal and 3d daily ozone. Each experiment consists of 9 ensembles with fully coupled ocean. It is found that interactive chemistry produces stronger stratospheric cooling, stronger strengthening of the polar night jet, and stronger poleward shift of the tropospheric jet, despite of the identical changes in ozone and shortwave heating rates. The authors attribute the difference to a chemistry-dynamics feedback that is absent when ozone is prescribed. This work highlights the importance to include the interactive chemistry into the climate simulations, and sheds light on understanding the complex coupling between the stratospheric ozone and the climate system. The paper is logically organized and well written. I am not fully convinced by the mechanisms the authors provided to explain the "chemistry-dynamics feedback", but I do think the results deserve publication. Below is my detailed comments:

1. The authors suggested that there is both positive and negative feedbacks between ozone changes and dynamics, which occurs at different seasons and levels, which involves the background zonal wind condition and wave-mean flow interaction. However, it is not clear why interactive chemistry and specified chemistry would behave different based on this mechanism. Both of them have the same changes in ozone and SW heating rates, then the initial changes in the zonal winds should also be similar since that simply follows the thermal wind balance. Wave-mean flow interaction would also work in a similar fashion in the two experiments.

In the specified chemistry setting, the ozone concentration is not dependent on the dynamics as it is the case for the interactive chemistry simulations. For example, the strength of the Brewer Dobson Circulation does not influence the ozone concentrations in Chem OFF, but it does in Chem ON. While a strong BDC in Chem ON would lead to a higher ozone concentration and therefore to a warming anomaly due to dynamics and ozone abundances, in Chem OFF only the dynamical warming would be guaranteed. By chance there might be also a higher ozone abundance but that is not necessarily the case since dynamics and chemistry are not coupled in Chem OFF. On the other hand, if there is an extreme high ozone year prescribed to Chem OFF, this will also be associated by a positive temperature anomaly due to radiative heating when sufficient solar energy is available, i.e. only the spring season (and early winter maybe) can effectively be affected by this. During such high ozone conditions, a cold winter vortex is still possible in Chem OFF, whereas unlikely in Chem ON.
We therefore think that the comparison of Chem ON and Chem OFF reveals feedbacks as discussed in the paper.

2. The mechanism for the "positive feedback" over the lower stratosphere in Nov/Dec is especially unclear, which is a key component to explain the stronger cooling seen in the interactive chemistry simulations. If the authors are referring to the positive

correlations in Fig. 3a, these positive correlations are very weak and not statistically significant.

Yes, this is the positive correlations we are talking about in the paper. When including the trend this feature is much stronger and more significant (Fig. A 3), but it is also apparent in the detrended Chem ON data. For the first version of the manuscript we decided to use significance hatching that indicates areas in which all of the single ensemble members show significant correlation coefficients. When considering significance when 7, 6 or 5 members out of the 9 members show significant correlation coefficients, the positive correlation patch increases in significance (Fig. A 4). Below we show these different cases (Figs. A 3 and A 4) as well as the correlation plots for the individual members (Fig. A 5). We changed the paper figure to the one showing significance when at least 5 members of the ensemble agree on a significant correlation coefficient.

[Figure]

*Figure A 3: Correlation for a) Chem ON and b) Chem OFF as in Figure 3 in the manuscript. In this case, the data was not detrended previously.*

We regard the positive correlation rather an indicator for a positive feedback than a validation since it shows the link between ozone and the dynamics and only indirectly includes the temperature response. We changed the wording in the manuscript where appropriate:

- Line 300: "Apart from the negative correlation also a positive correlation during stronger westerly background winds can be detected in Figure 3a in the lowermost stratosphere. It is less significant than the negative correlation but could be regarded as a hint for the positive feedback between ozone and the dynamical heating rates in Chem ON."
- Line 344: "A significant negative trend in the dynamical heating in the lowermost stratosphere during November and December is indicative of a positive feedback between ozone chemistry and the model dynamics (Lin et al. 2016), which is in agreement with the positive correlation in Figure 3a.

- Line 349: "[…], the maximum temperature trend in Chem OFF is weaker compared to Chem ON […] and occurs earlier, which could be due the lack of a positive feedback when ozone is prescribed rather than calculated interactively (compare Fig. 3)."

[Figure]

*Figure A 4: Correlation for Chem ON as in Figure 3 in the manuscript. Hatching-free areas indicate areas where a) 5 members b) 6 members and c) 7 members show significant correlation with a p-value <= 0.05.*

[Figure]

*Figure A 5:  Correlation for Chem ON as in Figure 3 in the manuscript but for each ensemble member separately. Hatching indicates p-value > 0.05.*

3. The climatology is calculated as the averaged over 1955-2013. But because of the strong trend related to ozone depletion, a large portion of the difference in the climatology may be a reflection of the difference in trends. It might be more meaningful to compare climatology over the pre-depletion era, such as 1955-1970.

We checked different periods for the calculation of the climatology. The climatological mean for the period 1955 to 1970 is shown in Figure A 6. The largest differences to whole period occur on the NH. On the SH, the significant differences between Chem ON and Chem OFF are weaker in SON before ozone depletion sets in. Nevertheless, since the ozone depletion period is the period of interest in the paper, we still use the whole period and include the figure for the pre-ozone hole period in the supplementary material now.
Line 308: "Since this period is strongly influenced by ozone depletion (see Suppl. Fig. 4 for a climatology of the pre-ozone hole period), […] ."

[Figure]

*Figure A 6: Climatological difference between Chem ON and Chem OFF as in Figure 1 of the paper but for the pre-ozone hole period 1955-1970 for a) zonal mean zonal wind at 10 hPa and b) zonal mean temperature at 30 hPa.*

4. Changes in the tropospheric jet. As shown in Fig. 9, the difference between Chem on and off is mainly between 20S-40S, which is more equator-ward than the equator flank of the tropospheric jet. I doubt the wind anomalies there can affect the location of the jet. Can the authors calculate the latitudes of the surface jets and verify if the latitude actually differ between the two? It is also not clear how the ozone anomalies in the polar regions affect the tropospheric jets in the subtropics.

We now include an additional figure about the tropospheric jet trend (latitude and amplitude) and find that the difference is not statistically significant as already implied by our previous plots. We are sorry for the over-interpretation of the previous differences for the tropospheric part and changed the according text passages.

5. Line 301-303: The first sentence here seems to suggest both the model used here and WACCM4 have stronger trends than WACCM-CCMI. But the second sentence suggests the opposite.

We rephrased the sentence:

Line 324: "The reduction in the trend from WACCM4 to WACCM-CCMI ca be […]"

Anonymous Referee #4

Review of "Sensitivity of the southern hemisphere tropospheric jet response to Antarctic ozone depletion: prescribed versus interactive chemistry" by Haase et al.

This paper investigates how Antarctic stratosphere and troposphere mean climate and climate change is affected by the model representation of Antarctic ozone. Three ensembles were performed for the 1955-2013 period using CESM with different ozone approach: interactive ozone, prescribed zonal-mean daily ozone, and prescribed 3D daily ozone. The results are consistent with previous studies that interactive ozone causes stronger Antarctic lower stratospheric cooling and stronger stratospheric jet response in austral summer.

This paper advances the understanding of the effects of interactive ozone on Antarctic climate change. Specifically, it emphasizes the role of reduced dynamical heating in causing Antarctic lower stratospheric cooling with interactive ozone. It also quantifies the impact of ozone-dynamics feedbacks and ozone zonal asymmetry on simulated temperature and jet trends. However, I think the authors' interpretation of how interactive ozone affects tropospheric jet trends needs to be clarified.

Major Comments:

I don't think the results presented in Figure 9 support the conclusion that interactive ozone leads to stronger poleward shift of the tropospheric jet. Figure 9 shows that interactive ozone does not significantly influence the tropospheric jet trends poleward of 40S. Significantly different tropospheric jet trends are only found between 20S and 40S (Fig. 9c), where the westerly trends are weaker in the interactive ozone simulations than in the prescribed ozone simulations. It would be useful to find out why interactive and prescribed ozone simulate different tropospheric subtropical jet trends.

However, the major point is that the large stratospheric circumpolar jet differences between interactive and prescribed ozone simulations do not propagate downward into the troposphere.

We agree with the referee and clarified the impact of interactive chemistry on the tropospheric jet. We kindly refer to the general comment to all referees at the beginning of this document for a more detailed answer.

Minor Comments:

-   The title should be changed because most of the paper is about the impact of interactive ozone on the SH stratosphere.

    We changed the title and do not refer to the tropospheric jet anymore.

- Lines 251-252: Why interactive ozone causes a weaker shallow BDC branch?

We think that the weaker shallow branch of the BDC is due to the fact the polar vortex is stronger in the Chem ON simulation leading to a reduced wave forcing and hence to a weakening of the BDC. Such a signature is evident in the shallow branch of the BDC (w* at 50 and 70 hPa in the Supplement). The response of the shallow branch could be due to the fact that this branch is faster reacting to changes in wave-breaking. To understand the process better, though, it might be necessary to investigate tropical upwelling. In our analysis, we did not consider the tropics so far.

- In some occasions, PNJ should be replaced with circumpolar jet.

We carefully went through the manuscript and changed PNJ to circumpolar jet where we thought it appropriate. See, for example lines 278, 283, and 303.

**Susan Solomon (Referee)**

This is an interesting and timely paper attacking an important problem. The findings are novel and certainly merit publication. I do have a number of questions and comments that I hope the authors find helpful in revising their paper. I don't think these suggestions are necessary since the paper is already quite good, but I do think they may make it clearer and stronger.

Substantive comments

1) The paper does a very good job on probing stratospheric change, but doesn't cover the tropospheric linkages as clearly. WACCM's Antarctic sea ice retreat has long been an issue in this (see Arblaster et al. recent paper) so it may be necessary to say that it's not a good model to study the problem, but linkages still should show up better in the data presented for comparison, and I am puzzled by that. Much more could be done (for example, do you think sea ice and surface temperature trends should be further discussed?), but at least what is shown should be clear. I am surprised that in Figure 5, IGRA was chosen for the temperature comparison; there are now rather better databases out there including ERA5 and MERRA2. I am also very surprised that the temperature trend does not penetrate into the troposphere in January in Fig 5 and 7, can you please discuss/explain. Since there are linkages seen in zonal winds shown in Figures 8 and 9, I am very puzzled. Perhaps it's down to choice of latitude range? Not clear to me. Also, such linkages are often clearer when geopotential height is plotted rather than temperature, and that might be considered. I think the paper needs a clearer bottom line on whether feedbacks matter or do not matter for the tropospheric response, or whether this model's poor simulation of sea ice changes means that it's not suitable for such testing and that the troposphere is therefore not the focus here.

Dear Susan, Thank you for your comments and suggestions! We were searching for the recent Arblaster paper you mentioned and came across the paper by Zoe Gillett from 2019 with Julie Arblaster as a second author (Gillett et al. 2019). We think this is the one you were talking about since it compares a fully-coupled WACCM version with a WACCM AMIP experiment. We contacted Zoe Arblaster and had a word about how important the ocean bias might be. We think that the ocean is not the crucial component and rather think that the discrepancy is due to the atmosphere since Zoe showed in her work that the impact of ozone onto the SAM was well captured in both model configurations (coupled and uncoupled ocean). The problem with the ocean bias is regarding the teleconnection to Australian temperatures, which was much better captured in the case that used observed SSTs. Also, a comparison between the older CAM4 and the more recent CAM5 model revealed that changes to the atmospheric parametrizations (from CAM4 to CAM5) improved the representation of this teleconnection pattern (while the ocean model did not change). We include the ocean now in the discussion

(line 496) but do not think that it is the main reason for the relatively weak coupling to the troposphere.

We chose IGRA since we were looking for an observational dataset that includes the period of 1969 to 1998. Especially during the first years of this period also ERA data is not very reliable. We think that the mean trend in IGRA is actually quite well captured especially compared to the results presented in Young et al. (2013). Figure A 7 shows the trends in polar cap temperature and zonal mean zonal wind for ERA data. There is also a gap between the stratosphere and the troposphere in the temperature trend. Maybe the connection is reduced by the gradual warming trend in the troposphere. This structure is similar in our WACCM simulations. To validate this assumption an ODS-only and GHG-only configuration would be useful. However, unfortunately we do not have these experiments available.

[Figure]

*Figure A 7: Polar cap temperature and zonal mean zonal wind trend in a combined dataset based on ERA-40 and ERA-Interim data. Significant trends are hatched.*

We agree that the bottom line of the paper needs to be clearer. This was also an issue that the other referees addressed. We now conclude that the impact on the troposphere is not very large (new analysis in the tropospheric jet trend shows this) and that a weak stratosphere-troposphere-coupling in WACCM in the SH might be the reason for that. The relatively weak coupling becomes obvious from the SAM timescale and was already shown in Gerber et al. (2010).

2) Total ozone is not a very sensitive diagnostic for evaluating a model's ability to simulate the coupling of interest here (Fig 4). Would it not be more useful to show some observations for comparison to Fig 4b instead. You could use the BDBP dataset or SWOOSH. It would be important to assess the vertical profile of the losses; much more so than the total column. Also, I would suggest that rather than showing the trends in ppmv per decade that you show the total trend over the period in percent. That is because what we really want to know is whether we see near complete depletion over the region from about 15-25 km at the peak, which gives us a good sense of the model's performance.

Figure 4 a was intended to mainly show the timing of the ozone trend in the lower stratosphere supporting the decision to investigate the 1969 to 1998 period. A comparison of Figure 4 b to observations is not as easy since observational data before the 1980s are very sparse. In Figure A 8, we show an example for the period 1979 to 2003. But since the validation of the ozone trend in WACCM is not the focus of our paper we decided to not include this comparison plot, also because it does not cover the period the paper investigates.

**1979 to 2003 polar cap ozone trend**

Figure A 8: Polar cap ozone trend in ERA-Interim and one member of the WACCM model with interactive chemistry.

3) It is well known that there is a heating rate issue in the way that the SC version of WACCM handles mesospheric ozone; in particular, diurnal effects are not correctly accounted for and there are spurious rates of heating as soon as you get up above about 2 mbar, where atomic oxygen and ozone interchange from day into night and drive a big diurnal variation that will give you an incorrect 24-hour average heating rate if you do not take account that SW heating is only present in daytime; this is all discussed in detail in Smith et al. Please comment on whether this may influence some of your results, particularly up near the 1 mbar level, and whether that has any potential to propagate downwards through a corresponding error in the residual circulation.

The heating rate issue is addressed as described in Smith et al. (2014) by prescribing also the total SW heating rates. There still is a bias in the upper stratosphere that might influence upper stratospheric processes to a certain degree but we are positive that this does not lead to spurious results in the lower stratosphere.

We include this information now in the manuscript to avoid confusion:
Line 200: "But we would like to mention that the ozone concentrations for the whole atmosphere and concentrations of other radiatively active species as well as the total short--wave heating rates above 65 km that are prescribed in the SC-WACCM simulations (Smith et al. 2014), are derived from the interactive chemistry WACCM simulations used in this study."

4) Lin et al. (J. Clim., 2009) showed evidence for a seasonal shift in the location of the polar vortex from July to November in the lower stratosphere. You have the perfect setup to test whether this occurs similarly irrespective of feedbacks and non-zonal forcing, and its relationship to BDC changes. It would be easy for you to reproduce their Figures 2-4, for your different cases; note the comparison to reanalysis shown in their Fig 8. It might help in understanding further what is going on in your Figure 11.

The Lin et al. 2009 paper is very interesting. Thank you or pointing it out! We did the analysis to reproduce Fig. 2 – 4 of Lin et al. (2009) for our different experiments. With one little difference – for simplicity we use the 50 hPa level. We include only a part of these figures here for the period discussed in our paper (1969 to 1998) and the full set of figures for the period Lin analyzed (1979 to 2007) in a separate PDF as a direct answer to your comment, since it is a lot of figures.

Because our analysis focuses on the December and January period, the results from Pu Lin's paper do not directly connect to our results. However, we included the paper as a reference. Line 448: "A wave-1 pattern in the lower stratospheric temperature trend was also described in Lin et al. (2009). They found that ozone cooling and dynamical warming were affecting different locations around Antarctica."

[Figure]

*Figure A 9: Chem ON, Eddy component of the climatology of T and O3 at 50 hPa for 1969 to 1998.*

As compared to Lin et al. 2009, the shift in location of the polar vortex is generally captured in WACCM. However, it is not as strong and delayed in time (Fig. A 9). In Lin et al. 2009 (Fig. 2) the strongest shift is found from September to November. A similar shift occurs in WACCM from October until December.

[Figure]

*Figure A 10: Chem ON, Trend of T and O3 at 50 hPa for 1969 to 1998. Hatching indicates insignificant trends at the 5% level, whereas single plots without any hatching are not significant at all.*

[Figure]

*Figure A 11: As Figure A 10, but for the Eddy component of the Trend of T and O3 at 50 hPa for 1969 to 1998.*

Regarding the patterns of the trend (Figs. A 10 and A 11), one can see that the eddy component is much weaker in WACCM as compared to the results shown in Lin et al. 2009. The shift from September to October shown in Lin et al. is not found in WACCM, which is probably due to fact that the ozone depletion in WACCM is centered over the pole rather than over East Antarctica.

In WACCM, the eddy components of the trends broadly resemble those of the climatology (Fig. A 9 and A 11). In Lin et al. a clear difference is found between the eddy components of the trends and the climatology. In that regard, CESM1(WACCM) does not seem to behave much better than the CMIP3 models analyzed in Lin et al. 2009.

The specified chemistry experiments show similar wave-1 patterns in the temperature climatology and trend patterns as compared to Chem ON with small differences in the position of the wave pattern. These figures are included in the additional PDF.

5) You emphasize the jet but it would be helpful to have more detail on the changes. It would be nice to make a simple scatter plot of the poleward shift of the jet (degrees) in the several simulations for various months, with the different simulations on the y axis and the observations on the x axis, or possibly using months on the x axis and plotting both data and models in the heart of the jet core on the y axis. We need to be able to see how many degrees the shift is in a simple way.

We now include an analysis for the tropospheric jet latitude and strength in the paper. We find that the trends in jet amplitude and latitude do not show strong differences between the different chemistry settings (new Figure 10 in the manuscript).

For your reference, Figure A 13 shows the climatological mean vs. trend for the jet latitude in the individual months of December, January and February for the model simulations and ERA data. One can see that significant trends occur mostly in January and December in the model and in reanalysis data. The modeled trend maximizes in January whereas the ERA trend maximizes in December. One can also see the offset between the climatological mean position of the tropospheric jet which is more than 5° too far south in WACCM.

[Figure]

*Figure A 12: Jet latitude trend versus climatological mean for ERA data (red), Chem ON (dark blue), Chem OFF (cyan) and Chem OFF 3D (yellow) for three different months individually (Dec - circle, Jan - triangle, Feb - square). Solid signs show significant trends following a Mann-Kandell Test.*

Minor comments

6) line 29 The paper's science is great but in several places the English is rather clumsy. Can this be checked over by a technical editor in one of the authors' institutions? Language like "The annually reoccurring depletion in polar stratospheric ozone was tremendous" distracts from an otherwise excellent paper. The annually reoccurring depletion in polar stratospheric ozone was striking or was of interest to scientists, the public, and policymakers alike, etc. would be better.

We revised the mentioned sentence as suggested and went through the manuscript carefully. I hope that we managed to improve the English in the manuscript. Unfortunately, we do not have a technical editor for this kind of issues. ACP, however, also includes an English prove

reading after acceptance of the manuscript, which hopefully helps with the most clumsy wordings.

7) line 30 Political action was taken to ban the responsible substances (termed: ozone depleting substances, ODSs) under the Montreal Protocol in 1987 is incorrect. Political action was begun that ultimately led to a ban on the responsible substances (termed: ozone depleting substances, ODSs) under the Montreal Protocol in 1987. The original Protocol in 1987 did not mandate a ban, only a freeze on emission at then-current rates.

Thank you for the comment. We corrected the sentence!

8) line 37 Need a reference to the first paper showing this by Shine (GRL, 1986).

We included the missing reference. Thank you pointing it out!

9) line 103 Please clarify what the shortcoming of Rae et al. is; this is not very clear here. How does it work and why doesn't it capture heterogeneous loss?

The aim of their method is to generate a zonally asymmetric ozone field that is consistent with the model dynamics from a prescribed zonal mean ozone field while maintaining the climatological zonal mean as prescribed. They use potential vorticity asymmetries to generate an asymmetry-scaling coefficient factor to apply to the prescribed zonal mean ozone. Since this method depends on the spatial pattern of PV, heterogeneous loss is not directly covered. So, only when ozone is following the dynamics, the method works properly. For more details we have to refer to the study of Rae et al. (2019).

We revised the section in the manuscript as follows:
Line 111: "Also worth mentioning is Rae et al. (2019), who designed a computationally efficient method to interactively re-scale prescribed ozone values to a dynamically model--consistent 3D ozone field based on the potential vorticity field of the model. This method, unfortunately, is not well suited to represent the observed SH ozone depletion since it follows a solely dynamical approach and has therefore difficulties to account for heterogeneous chemistry processes."

[revised manuscript text omitted]